# OCTAVIUS: MITIGATING TASK INTERFERENCE IN MLLMS VIA LORA-MOE

**Zeren Chen**[1,2*], **Ziqin Wang**[1,3*], **Zhen Wang**[2], **Huayang Liu**[2]
**Zhenfei Yin**[1,4†], **Si Liu**[3], **Lu Sheng**[2‡], **Wanli Ouyang**[1,4], **Jing Shao**[1‡]
[1] Shanghai AI Laboratory, [2] School of Software, Beihang University,
[3] Institute of Artificial Intelligence, Beihang University, [4] University of Sydney
`{czr1604,wzqin,wz1234,hy43,liusi,lsheng}@buaa.edu.cn`,
`{yinzhenfei,shaojing}@pjlab.org.cn, wanli.ouyang@sydney.org.cn.`

## ABSTRACT

Recent studies have demonstrated Large Language Models (LLMs) can extend their zero-shot generalization capabilities to multimodal learning through instruction tuning. As more modalities and downstream tasks are introduced, negative conflicts and interference may have a worse impact on performance. While this phenomenon has been overlooked in previous work, we propose a novel and extensible framework, called Octavius, for comprehensive studies and experimentation on multimodal learning with Multimodal Large Language Models (MLLMs). Specifically, we combine the well-known Mixture-of-Experts (MoE) and one of the representative PEFT techniques, *i.e.,* LoRA, designing a novel LLM-based decoder, called LoRA-MoE, for multimodal learning. To the best of our knowledge, we are one of the pioneering efforts to introduce MoE into MLLMs to address this problem. The experimental results (about 20% improvement) have shown the effectiveness and versatility of our design in various 2D and 3D downstream tasks. Code and datasets are available at `https://openlamm.github.io/paper_list/Octavius`.

## 1 INTRODUCTION

Multimodal Large Language Models (MLLMs) (Alayrac et al., 2022; Huang et al., 2023; Liu et al., 2023; Li et al., 2023a; Zhu et al., 2023) have been considered as promising general-purpose interfaces that can perform various multimodal tasks under few-/zero-shot settings. Apart from leveraging the powerful Large Language Models (LLMs) (OpenAI, 2023; Touvron et al., 2023a) as the universal interfaces that unify the responses to different types of tasks as task-specified textual sequences, the keys to the success of MLLMs are to reliably perceive more modalities and be efficiently fine-tuned to adapt more downstream tasks.

To achieve this goal, MLLMs rely on the instruction-tuning scheme (Ouyang et al., 2022) where the model is fine-tuned based on multimodal instruction-following dialogues orchestrated from various multimodal tasks. Moreover, thanks to the Parameter-Efficient Fine-Tuning (PEFT) techniques (*e.g.*, LoRA (Hu et al., 2021) and Adapter (Houlsby et al., 2019)) where only small trainable components are injected in the model and updated during fine-tuning, recent MLLMs (Zhang et al., 2023; Yin et al., 2023; Ye et al., 2023) can efficiently learn to solve downstream tasks with a small scale of annotated data, while preserve the language proficiency and generalizability to novel situations. Remarkably, these models achieve comparable performance at low costs in comparison to LLaVA (Liu et al., 2023), KOSMOS series (Huang et al., 2023; Peng et al., 2023) and Shikra (Chen et al., 2023), which are learned by full model fine-tuning with a large amount of multimodal data.

However, PEFT has to address the crucial tug-of-war problem (Hadsell et al., 2020), where simultaneously learning different tasks may cancel each task-specific optimization out, and ultimately com-

---

[*]Equal contribution.
[†]Project leader.
[‡]Corresponding author.

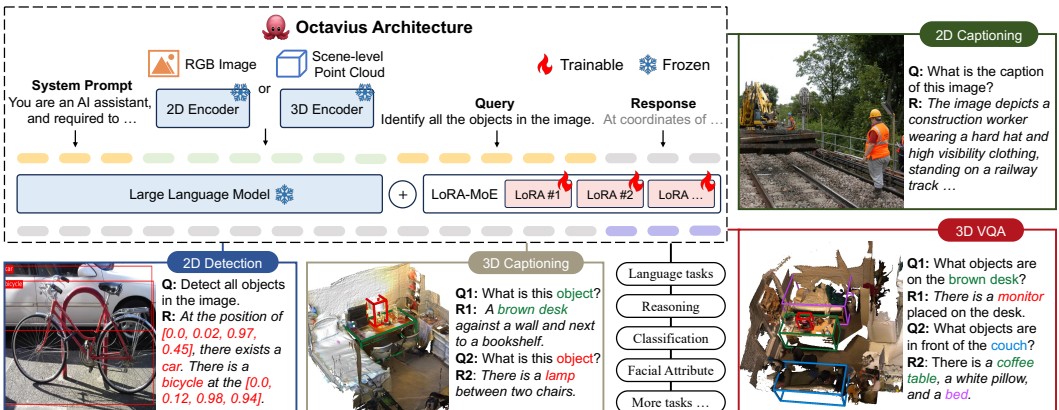

Figure 1: **Octavius** is a unified, multimodal large language model with a novel capability to comprehend various tasks across different modalities, including but not limited to 2D captioning, 2D detection, 3D VQA, and 3D dense captioning.

promise the performance of each downstream task. This problem is much more severe in MLLMs, especially when more modalities and tasks are involved, but only a few well-annotated data are available. First, the features from new modalities are not easy to be aligned with each other, not to mention compatible with the LLM-based language decoders. Second, simultaneously learning to acquire knowledge at distinct granularities, such as the instance-level perception (*e.g.*, object detection) and logical reasoning (*e.g.*, VQA), may lead to significant interference. Third, it is more complicated by using an LLM-based decoder to generate textual responses that meet the special requirements of tasks in modalities other than natural language, such as the bounding box coordinates in detection tasks, or action sequences for robotics.

To resolve this issue, we propose **LoRA-MoE**, which combines the well-known Mixture-of-Experts (MoE) (Jacobs et al., 1991; Jordan & Jacobs, 1994) and one of the representative PEFT techniques, *i.e.*, LoRA (Hu et al., 2021). Based on LoRA-MoE, an LLM-based decoder can efficiently be involved in more downstream tasks and more modalities by learning more LoRA modules. Different from conventional MoE models (Shazeer et al., 2017; Lepikhin et al., 2020; Fedus et al., 2022; Du et al., 2022), we adopt a simple yet effective instance-based gate routing scheme, sparsely activating independent LoRA experts with instance-level instructions and further acquiring task-specific knowledge for better aligning different tasks. Notably, to the best of our knowledge, we are one of the pioneering efforts to introduce MoE into MLLMs to address the tug-of-war problem.

To validate the effectiveness of LoRA-MoE, in this work, we investigate a more complicated scenario, where the MLLMs should simultaneously learn downstreaming tasks from more additional modalities, such as 2D images and 3D point clouds. This scenario is especially useful for embodied agents (Duan et al., 2022; Mu et al., 2023; Driess et al., 2023). Specifically, in addition to the off-the-shelf image encoder, we design a point cloud encoder called **Object-As-Scene**, which provides language-aligned scene-level point cloud representations. This encoder at first gathers language-aligned point cloud features of each instance (Xue et al., 2023) in a scene, which are then aggregated into a scene-level feature based on the attention operation guided by the input instructions.

Based on the aforementioned contributions, we introduce a novel and extensive framework called **Octavius**, which learns the MLLMs upon the instruction-following datasets adapted from LAMM (Yin et al., 2023) and ScanNet (Dai et al., 2017). As shown in Figure 1, Octavius can successfully address various 2D/3D vision and language tasks, including but not limited to 2D detection, 2D captioning, 3D VQA, and 3D dense captioning. We conduct various experiments to validate the effectiveness and versatility of our design, improving multiple downstream tasks by about 20% while increasing only a few trainable parameters.

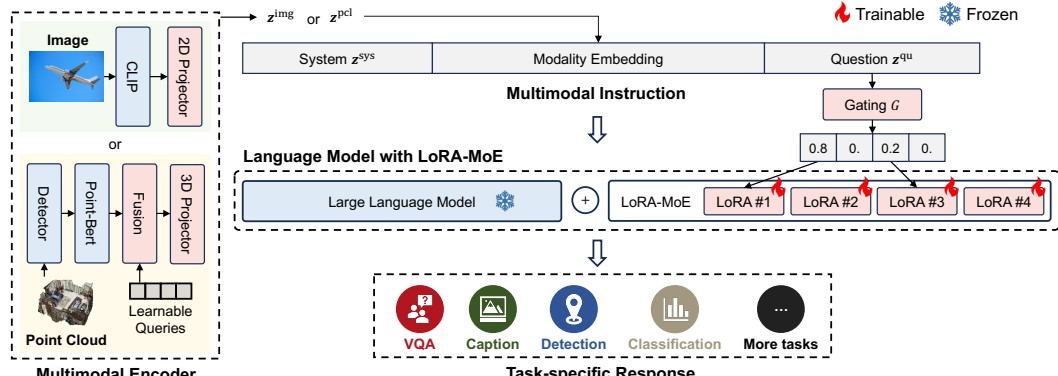

Figure 2: **Overall pipeline of Octavius**. We design corresponding encoders for different modalities, with the primary objective of empowering the LLMs to gain a deeper understanding of visual features. Additionally, we propose a dynamic gating network that selects distinct LoRA experts based on input instructions, thereby proficiently mitigating interference arising from multimodal learning.

## 2  RELATED WORKS

**Large Language Models (LLMs) & PEFT.** Recently, Large Language Models (LLMs) (Brown et al., 2020; Chowdhery et al., 2022; Chiang et al.; Touvron et al., 2023b) have gained significant attention due to their impressive capabilities in language generation (Zhang et al., 2022a), in-context learning (Wei et al., 2022), and reasoning (Touvron et al., 2023a). For both data- and compute-efficient adaptation on certain downstream tasks, several PEFT (Parameter-Efficient Fine-Tuning) (Li & Liang, 2021; Houlsby et al., 2019; Karimi Mahabadi et al., 2021; Hu et al., 2021) are proposed. For instance, LoRA (Hu et al., 2021) represents weight updates using two smaller matrices through low-rank decomposition, where original weights are kept frozen while the new update matrices are trained. In this work, we adopt LoRA for efficient MLLMs fine-tuning.

**Multimodal Large Language Models (MLLMs).** Several recent studies have attempted to extend the capability of LLMs to multimodal tasks. Alayrac et al. (2022); Li et al. (2023b); Liu et al. (2023); Zhang et al. (2023); Yin et al. (2023); Chen et al. (2023); Peng et al. (2023) introduce image modality in LLMs for comprehending 2D visual content. Hong et al. (2023) combines LLMs with 3D modality by rendering point clouds into 2D images and utilizing them to represent 3D visual features. Driess et al. (2023); Mu et al. (2023); Brohan et al. (2023) establish connections between visual inputs and embodied controls for robotic tasks. Despite its wide range of multimodal applications, the performance degradation caused by interference between tasks and modalities during fine-tuning in MLLMs receives inadequate attention.

**Mixture-of-Experts (MoE).** Deep MoE models are proposed to increase the number of model parameters without adding computational overhead in the field of computer vision (Riquelme et al., 2021; Mustafa et al., 2022; Shen et al., 2023) and natural language processing (Shazeer et al., 2017; Lepikhin et al., 2020; Fedus et al., 2022). Different from these approaches, we aim to address conflicts between tasks with MoE. Adamix (Wang et al., 2022a), an approach related to but distinct from ours, randomly selects experts during training and uses the average weights of experts in inference, which may be analogous to dropout (Srivastava et al., 2014) in certain cases. In this paper, we desire for a dynamic gate routing strategy to automatically calculate the weights of each LoRA expert according to the input instructions, adapting MLLMs for broader multimodal applications.

## 3  METHODOLOGY

As illustrated in Figure 2, we propose an extensible framework called **Octavius** for multimodal instruction tuning. In Section 3.1, we first elaborate on the tug-of-war problem and propose a unified LoRA-MoE decoder to break through the bottleneck caused by interference between different tasks and modalities. We then verify our design using both image and point cloud modality in this work and describe corresponding encoders in Section 3.2.

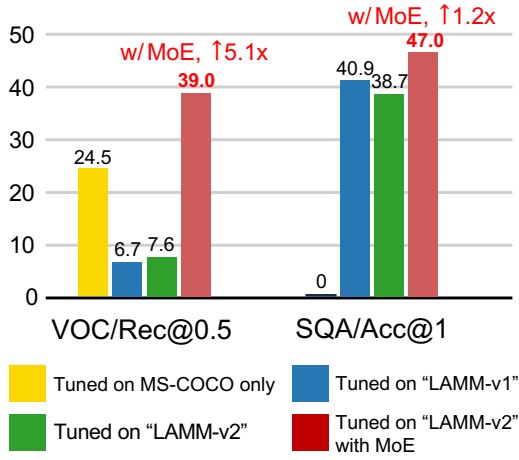

Figure 3: We conduct a simple pilot study on PASCAL VOC and ScienceQA to demonstrate the tug-of-war problem and the effectiveness of our proposed LoRA-MoE. Recall@0.5 denotes recall at an IoU threshold of 0.5, respectively.

Figure 4: We follow previous works, *e.g.*, LAMM (Yin et al., 2023), LLaVA (Liu et al., 2023), to apply an instruction-following training pipeline.

## 3.1 MULTIMODAL DECODER

### 3.1.1 THE TUG-OF-WAR PROBLEM

Interference among different modalities and tasks is a common and critical issue (Zhao et al., 2018; Vandenhende et al., 2021) in multimodal and multitask learning. While MLLMs can alleviate this problem by adopting the same learning objective, *i.e.*, next-token prediction loss, for all tasks, there still exists task-specific divergences that limit their potential in various downstream tasks. Since previous works have yet to delve into the tug-of-war phenomenon in MLLMs, we conduct a simple pilot study on image modality to reveal this problem.

The pipeline of image encoders in MLLMs are simple and similar in previous works. Here, we select LAMM (Yin et al., 2023) as our model due to its rich benchmarks on downstream tasks. We fine-tune LoRA and projector in LAMM following (Yin et al., 2023) and validate zero-shot performance on PASCAL VOC (Everingham et al., 2010) and ScienceQA (Lu et al., 2022) datasets. We report the recall and precision at an Intersection over Union (IoU) threshold of 0.5 on PASCAL VOC and the accuracy of multiple-choice questions on ScienceQA.

The results are shown in Figure 3. Although the original dataset of LAMM, referred to as "LAMM v1", contains numerous images from MS-COCO (Lin et al., 2014), the lack of sufficient detection instructions results in poor performance on PASCAL VOC. To overcome this problem, we leverage the entire COCO detection annotations and GPT-API (OpenAI, 2023) to generate additional detection instructions as supplementation, constructing a new dataset called "LAMM v2" for better generalization of detection tasks. After verifying the detection ability of LAMM by using COCO detection instructions alone, we find using a mixed dataset does not lead to a huge improvement in detection performance. Also, there is a decline in the VQA tasks. Moreover, we can achieve the same results on another dataset used in LLaVA (Liu et al., 2023). It can be concluded that MLLMs suffer from a severe tug-of-war problem on image modality, not to mention incorporating more modalities for training simultaneously.

### 3.1.2 LORA-MOE

Some prior works (Kendall et al., 2018; Chen et al., 2018) have attempted to balance the magnitudes of losses or gradients across different tasks to address the tug-of-war issue. However, considering that different objectives are defined for each task in multitask learning, it is challenging to directly extend existing methods to MLLMs that adopt a unified optimization objective for all tasks. In this section, we introduce the concept of Mixture-of-Experts (Jacobs et al., 1991; Jordan & Jacobs,

1994; Shazeer et al., 2017), proposing a unified LoRA-MoE decoder based on a instance-based gate routing strategy.

**Revisiting MoE Models.** A typical MoE Model injects multiple MoE layers into LLM to accommodate a greater number of parameters. The MoE layer consists of a group of $N$ expert networks $E_1, E_2, ..., E_N$ and a gating network $G$, taking the previous tokens as input and producing the probability of the next token:

$$\texttt{tok}_i = \sum_k^N G(\texttt{tok}_{0...i-1})_k E_k(\texttt{tok}_{0...i-1}), \tag{1}$$

where $\texttt{tok}_i$ denotes $i$-th token. We refer to this kind of gating network, which based on token-level input, as **token-based gate**. Furthermore, to prevent $G$ from consistently producing imbalanced weights that favor only a few experts, an auxiliary loss $\mathcal{L}_{\text{balance}}$ (Shazeer et al., 2017; Zhou et al., 2022; Wu et al., 2022) is introduced to balance gating routing. For example, Shazeer et al. (2017) minimize the coefficient of variation of the gate values for each token, encouraging all experts to have equal importance:

$$\mathcal{L}_{\text{balance}} = \alpha \, \texttt{CV}\bigg[\sum_i G(\texttt{tok}_{0...i-1})\bigg]^2, \tag{2}$$

where $\alpha$ is a hyper-parameter and $\texttt{CV}(\cdot)$ is the coefficient of variation, which is the ratio of the standard deviation to the mean.

**LoRA-MoE and Instance-based Gate Routing.** Different from token-based gate in LLM, we design a simple but effective routing strategy for MLLMs, assigning downstream tasks to independent experts for specific knowledge based on individual instances, called **instance-based gate**. It is motivated that the input questions $z^{\text{qu}}$ applied in multimodal instructions will substantially affect the responses generated by MLLMs, we take the questions as input to predict routing scores for each expert. Then, we select sparsely-activated experts based on routing scores for each individual instance to generate the entire sentence. In this work, the LoRA module is treated as an expert in MLLMs, combining instance-based gate with it to alleviate interference arise from multimodal learning, named **LoRA-MoE**. By replacing LoRA in each projection layer of language model $f^{\text{LLM}}$ with a group of independent LoRA experts $\{E^{\text{LoRA}}\}_N$, we can predict the $i$-th token value as follow:

$$\texttt{tok}_i = f^{\text{LLM}}(\texttt{tok}_{0...i-1}) + \sum_k^N G(z^{\text{qu}})_k E_k^{\text{LoRA}}(\texttt{tok}_{0...i-1}). \tag{3}$$

Additionally, we find that $\mathcal{L}_{\text{balance}}$ is incompatible and infeasible with LoRA-MoE in an instance-based gate scenario. For example, it is more reasonable to assign detection samples to a LoRA expert proficient in localizing than the other experts for the purpose of balancing. Therefore, we can observe some imbalance phenomenon in experiments (see Section 4.3 for details), unless the amount of data for each task in the whole dataset is balanced.

Compared with previous MoE models, LoRA-MoE allows for efficient fine-tuning on small datasets and faster convergence with instance-based gate routing. During the inference phase, if the downstream tasks and input questions are specified, LoRA-MoE can also merge parameter weights with language model like vanilla LoRA to reduce storage requirements and inference costs.

**Instruction Tuning with LoRA-MoE.** Given the target modal features $z^{\text{img}}$ or $z^{\text{pcl}}$, we construct image-text conversation pairs in an instruction-following format based on previous works (Zhang et al., 2023; Yin et al., 2023; Peng et al., 2023), as shown in Figure 4. The language model with LoRA-MoE is then trained to predict corresponding responses based on the system prompts, target modal features and questions.

## 3.2 MODALITY ENCODER

### 3.2.1 IMAGE ENCODER

Benefit from the pioneer vision-language model (Radford et al., 2021) that bridges the gap between the image and language modality, Li et al. (2023a); Liu et al. (2023); Chen et al. (2023); Zhang et al.

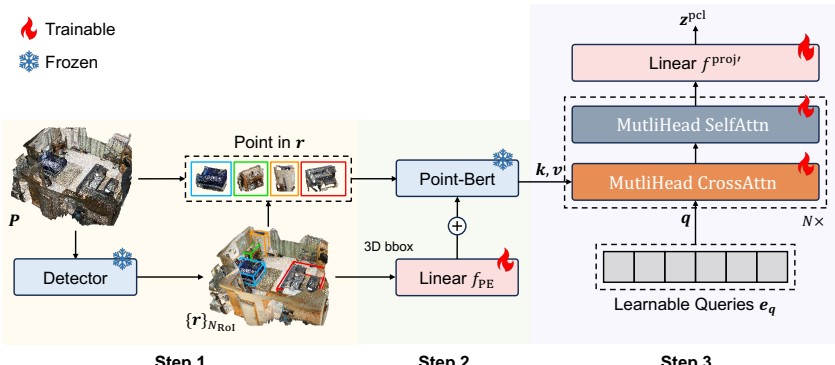

Figure 5: **Structure of Object-As-Scene**. To acquire scene-level features, we follow a three-step process. Firstly, we obtain RoIs from a given point cloud using a pre-trained detector. Next, we pre-train a Point-Bert model following a ULIP-like pipeline and employ it to extract instance-level 3D features. Finally, by aggregating features from visual embedding, we derive the final scene-level feature.

(2023); Yin et al. (2023) achieve impressive results. We follow their pipeline to extract features for image modality. Specifically, for an image input $I \in \mathbb{R}^{H \times W \times 3}$, we use the pre-trained CLIP visual encoder ViT-L/14 $f^{\text{CLIP}}$ (Radford et al., 2021) to extract the language-aligned visual feature $h^{\text{img}}$, following by a trainable linear layer $f^{\text{proj}}$ to match the dimension of $h^{\text{img}}$ with the word embedding space in language model:

$$h^{\text{img}} = f^{\text{CLIP}}(I); \ z^{\text{img}} = f^{\text{proj}}(h^{\text{img}}), \tag{4}$$

where $z^{\text{img}}$ is the output features of image modality for further instruction tuning.

### 3.2.2 POINT CLOUD ENCODER

Conventional 3D methods typically apply 3D CNNs (Yan et al., 2018; Shi et al., 2020; Qi et al., 2017) or Transformers (Zhao et al., 2021; Fan et al., 2022) as feature extractors to process sparse point cloud data. However, they still retain numerous background points with low-density information, which may confuse the subsequent language models in MLLMs, ignoring the pivotal elements in the scene. Besides, the unavailability of encoders capable of aligning scene-level 3D features with language may pose significant challenges for LLMs in comprehending the semantic information of the entire scene. To address these issues, we propose Object-As-Scene as our point cloud encoder dedicated to language-aligned scene-level 3D representation generation, as illustrated in Figure 5.

**Step 1: Locating Regional RoIs as Candidates.** An intuitive way to avoid excessive background points is to identify specific regions in the scene that may contain instances or relevant semantic information and encode entire scene with these regions. Specifically, given a 3D point cloud scene, we employ a pre-trained object detector, *i.e.*, FCAF3D (Rukhovich et al., 2022), to locate candidate RoIs (Region-of-Interest) $\{r\}_{N_{\text{RoI}}}$. Note that $N_{\text{RoI}}$ denotes the number of RoIs. Besides, for tasks such as captioning or classification that primarily focus on instances mentioned in the conversations, we directly use regional features associated with these instances as input.

**Step 2: Extracting RoI Features Aligned with Language and Image.** Inspired by a recent work (Xue et al., 2023), we pre-train a Point-Bert (Yu et al., 2022) encoder $f^{\text{Point-Bert}}$ aligned with both language and image modalities following a ULIP-like pre-training pipeline, allowing us to extract instance-level 3D visual features from points $P \in \mathbb{R}^{N \times 6}$ in candidate RoIs $\{r\}_{N_{\text{RoI}}}$ during instruction tuning:

$$\{h^{\text{pcl}}\}_{N_{\text{RoI}}} = f^{\text{Point-Bert}}(P, \{r\}_{N_{\text{RoI}}}). \tag{5}$$

More details about improved ULIP-like pre-training pipeline can be found in the Appendix.

**Step 3: Aggregating RoI Features as Scene.** Next, we adopt a fusion module with two stacked transformer layers to fuse scene-level features with RoI features. Specifically, we utilize multi-head cross-attention mechanism (Vaswani et al., 2017) (denoted as MHCA) to attend a group of trainable

Table 1: **Comparisons on image modality.** We investigate the zero-shot performance of our method on multimodal (2D) scenarios. We use the following abbreviations in this table and subsequent experiments for ease: "Det." for detection tasks, "Cap." for captioning task, and "Cls." for classification tasks.

| Models | MoE | FT. Dataset | Det. (IoU=0.5) | | VQA | Cap. | Cls. | Facial Attr | | Avg. |
|---|---|---|---|---|---|---|---|---|---|---|
| | | | Recall | Prec | Acc@1 | CIDEr | Acc@1 | Hair Acc@1 | Smile Acc@1 | |
| LAMM | | LAMM v2 | 7.61 | 5.95 | 40.31 | 0.21 | **73.50** | 58.04 | 50.15 | – |
| | ✓ | | **39.04** | **35.21** | **46.95** | **5.66** | 65.40 | **60.93** | **59.82** | 20.89% ↑ |
| LLaVA-LoRA | | LLaVA | – | – | 52.35 | **30.75** | 2.89 | **12.50** | 50.23 | – |
| | ✓ | | – | – | **55.58** | 23.08 | **41.00** | 3.93 | **52.17** | 18.36% ↑ |

queries $e_q$ with RoI features $\{h^{\text{pcl}}\}_{N_{\text{RoI}}}$:

$$h_q^{\text{pcl}} = \text{MHCA}(q = e_q, kv = \{h^{\text{pcl}} + f_{\text{PE}}(r)\}_{N_{\text{RoI}}}). \qquad (6)$$

$f_{\text{PE}}$ is used to transform 3D bbox coordinates into positional embedding to enrich spatial information. And $q, k$ and $v$ denote query, key, and value in attention. As in image encoder (Section 3.2.1), a trainable linear layer is also applied for final 3D features $z^{\text{pcl}}$:

$$z^{\text{pcl}} = f^{\text{proj}'}(h_q^{\text{pcl}}) \qquad (7)$$

**3D Instruction Data.** We construct a 3D instruction tuning dataset called **Scan2Inst** using ScanNet (Dai et al., 2017) as our 3D instruction tuning dataset due to its diverse tasks and annotated categories. Following Wang et al. (2022b), we use GPT-API to generate a total of 80k data pairs comprising instructions and responses based on original dataset.

## 4 EXPERIMENT

### 4.1 EXPERIMENT SETUP

To explore the effectiveness of our framework in multimodal learning, we fine-tune Octavius in three modality setups: **i.)** image modality, **ii.)** point cloud modality and **iii.)** both image and point cloud modalities. We then evaluate the zero-shot performance using these three fine-tuned models on various downstream tasks. More details about architecture and training scheme are provided in Appendix.

**Instruction Datasets.** For image modality, we follow Yin et al. (2023) to construct "LAMM v2", an instruction dataset consisting of MS-COCO (Lin et al., 2014) and Bamboo (Zhang et al., 2022b), which includes object detection, classification, captioning, and other common 2D tasks. For point cloud modality, we utilize ScanNet (Dai et al., 2017) to generate an instruction dataset called "Scan2Inst" which contains VQA, captioning, and classification tasks. In the multimodal learning (2D&3D) setup, we merge the image and point cloud instruction dataset to fine-tune the entire framework simultaneously.

**Quantitative Zero-shot Evaluation.** We perform zero-shot evaluation on various downstream tasks for both image and point cloud modalities. For image modality, we perform Visual Question Answering (VQA) on ScienceQA (Lu et al., 2022), classification on CIFAR-10 (Krizhevsky et al., 2009), captioning on Flickr30K (Young et al., 2014) and facial attribute recognition on CelebA (Liu et al., 2018). Note that we evaluate performance in VQA tasks through multiple-choice selection. For point cloud modality, we perform classification on ShapeNet (Chang et al., 2015) (55 classes) and captioning on NR3D (Achlioptas et al., 2020). We also evaluate the performance of classification, captioning, and QA on the test split of ScanNet to verify the proposed Object-As-Scene encoder.

### 4.2 QUANTITATIVE RESULTS.

All results are provided in Table 1, 2, 3 for different modality setups. Severe interference can be found in the experimental results, especially between the localizing tasks and semantic understanding tasks like VQA and captioning. After equipping with LoRA-MoE, we can observe a remarkable

Table 2: **Comparisons on point cloud modality.** We report both Fine-Tuning (FT.) Results and Zero-shot (ZS.) Results on the 3D tasks. We also compare our model with 3D-LLM (Hong et al., 2023). Here, $^\dagger$ indicates the results of Scan2Cap are evaluated on a custom test set regenerated by 3D-LLM, which is different from ours.

| Models | FT. Results | | | | | ZS. Results | | | ZS. Avg. |
|---|---|---|---|---|---|---|---|---|---|
| | Cap. (Scan2Cap) | | VQA (ScanQA) | | Cls. (ScanNet) | Cls. (ShapeNet) | Cap. (Nr3d) | | |
| | BLEU-1 | CIDEr | BLEU-1 | CIDEr | Acc@1 | Acc@1 | BLEU-1 | CIDEr | |
| 3D-LLM (Flamingo) | 36.10$^\dagger$ | – | 30.30 | 59.20 | – | – | – | – | – |
| Ours | 33.58 | 35.11 | 43.21 | **168.21** | 47.40 | 19.75 | 20.02 | 16.19 | – |
| Ours w/ MoE | **35.94** | **39.38** | **44.24** | 167.31 | **48.80** | **24.85** | **21.16** | **17.22** | 17.06% ↑ |

Table 3: **Comparison on image & point cloud modalities.** While there are some performance gaps compared to the model fine-tuned on a single modality, our model with the inclusion of MoE exhibits a superior performance ($\sim$20%) compared to its counterpart.

| FT. Dataset | MoE | 2D Results (ZS.) | | | | | | 3D Results (FT.) | | | 3D Results (ZS.) | | Avg. |
|---|---|---|---|---|---|---|---|---|---|---|---|---|---|
| | | Det. | VQA | Cap. | Cls. | Facial | | Cap. | VQA | Cls. | Cls. | Cap. | |
| | | Rec@0.5 | Acc | CIDEr | Acc | Hair | Smile | CIDEr | CIDEr | Acc | Acc | CIDEr | |
| LAMM v2 | | 7.61 | 40.31 | 13.28 | 73.50 | 58.04 | 50.15 | – | – | – | – | – | – |
| | ✓ | 39.04 | 46.95 | 26.71 | 65.40 | 60.93 | 59.82 | – | – | – | – | – | – |
| Scan2Inst | | – | – | – | – | – | – | 39.56 | 162.14 | 47.60 | 19.75 | 16.19 | – |
| | ✓ | – | – | – | – | – | – | 39.38 | 167.31 | 43.40 | 24.85 | 17.22 | – |
| LAMM v2+Scan2Inst | | 2.64 | **39.71** | 0.04 | **71.66** | 42.47 | 50.66 | 19.76 | **182.00** | 38.80 | 14.85 | 8.26 | – |
| | ✓ | **34.30** | 35.80 | **10.06** | 56.86 | **51.52** | **54.22** | **33.29** | 181.44 | **47.20** | **21.10** | **17.22** | 21.40% ↑ |

improvement of approximately 20% in all setups, demonstrating the effectiveness of our design in resolving the tug-of-war issue. Additionally, we compare our proposed point cloud encoder Object-As-Scene with a recent work, 3D-LLM (Hong et al., 2023) in Table 2. We achieve a comparable performance on Scan2Cap, and outperform 3D-LLM on ScanQA by a significant margin, suggesting a better scene-level understanding capability of Object-As-Scene. Besides, as shown in Table 3, as the complexity of interference among tasks increases, especially when tasks of different modalities are introduced, we can observe a huge performance drop when training the model with different modalities like image and point cloud simultaneously compared to separate training. Our proposed MoE-based decoder partially alleviates the degradation and even achieves comparable performance with separate training in some tasks.

## 4.3 ABLATION AND ANALYSIS

**LoRA-MoE.** The results are shown in Table 4. We first ablate on MoE architecture by individually employing dedicated LoRAs for each tasks (denoted as "Individual" in the table). While the individual gate exhibits performance merits in specific tasks, its primary challenge is the difficulty in assigning suitable experts for tasks that are not encountered in the instruction dataset, thereby compromising the model's generalizability. Another observation is the superior efficacy of the sparse gate relative to the dense gate (denoted as "Weighted Sum" in the table), which is intuitive when considering that the dense gate can essentially be regarded as a singluar LoRA with additional parameters. Furthermore, we compare the performance of the sparse gate against the baseline model (single LoRA) under conditions of parameter-consistency during inference (*i.e.*, sparse top-2 gate only uses half of rank in LoRA compared with baseline model). The enhanced performance of sparse gate demonstrate that **MoE transcends a mere aggregation of parameters**.

**Gate Routing in MoE and Load Balancing.** As shown in Figure 6, there is a huge discrepancy in expert selection between detection and VQA tasks, which demonstrates the tug-of-war phenomenon, and explains why using a single LoRA yields poor performance on both tasks simultaneously. Additionally, it is found that the routing weights of experts assigned by gate network tend to concentrate on a subset of specific experts. In particular, in a 4-expert model, despite the superior performance compared to a 3-expert model, the final converged model ends up utilizing only 3 of 4 available experts. To further explore this imbalance issue, we conduct several experiments with load balancing loss (Equation 2) in Table 5. As a result, no improvements and better routing results are

Table 4: **Ablation studies on MoE architecture on 2D tasks.** "Sparse Top-2" gate picks out top-2 ranked experts based on routing scores. "Weighted Sum" gate uses the weighted sum of all experts as output. "Individual" gate employs different experts for each 2D tasks individually. "#Trainable Param." denotes the proportion of trainable parameters to total parameters.

| Gate Type | LoRA-Rank | Det. (VOC, IoU=0.5) | | VQA | #Trainable Param. |
| | | Recall | Prec. | Acc@1 | |
|---|---|---|---|---|---|
| – (Baseline) | 32 | 7.61 | 5.95 | 40.31 | 0.4% |
| Sparse Top-2 | 32 | 39.04 | 35.21 | 46.95 | 1.6% |
| Weighted Sum | 32 | 9.78 | 5.33 | 44.71 | 1.6% |
| Individual | 32 | 28.38 | 25.64 | 48.54 | 2.4% |
| Sparse Top-2 | 16 | 32.81 | 24.46 | 39.11 | 0.8% |
| Sparse Top-2 | 8 | 25.44 | 21.87 | 37.65 | 0.4% |

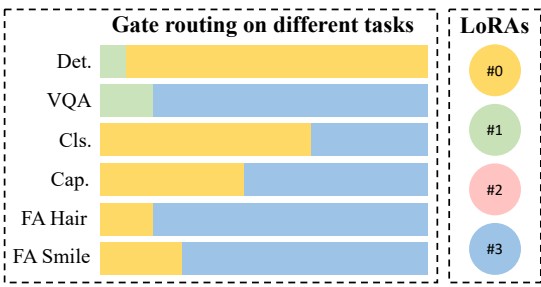

Figure 6: **Gate routing on different 2D tasks.** We use different color to represent $i$-th LoRA. The proportion represents the score of each expert.

Table 5: **Ablation studies on load balancing in MoE.** "LB" means apply load balance loss. **Token** means using token-based gate.

| #Experts | LB | Token | Det. | | VQA |
| | | | Recall | Prec. | Acc@1 |
|---|---|---|---|---|---|
| 4 | | | 39.04 | 35.21 | 46.95 |
| 4 | ✓ | | 33.21 | 26.80 | 45.26 |
| 8 | | | 22.30 | 11.01 | 39.91 |
| 8 | ✓ | | 21.52 | 12.10 | 37.53 |
| 4 | | ✓ | Fail | | |
| 4 | ✓ | ✓ | Fail | | |

observed, which is intuitive because directly employing load balancing strategies is incompatible and infeasible in an instance-based gate scenario (see Section 3.1.2). We also attempt to replace the instance-based gate with token-based gate used in conventional MoE methods (Shazeer et al., 2017; Lepikhin et al., 2020). However, since only the LoRA modules are trained in our approach, the number of trainable parameters differs from conventional MoE models where all parameters of the entire foundation model are trained, resulting in poor convergence.

Besides, we also provide more ablations and analysis on point cloud encoder, LoRA-MoE, gate routing and qualitative results in Appendix.

## 5 CONCLUSION AND LIMITATIONS

In this paper, we propose Octavius, a unified multimodal framework, to effectively address the critical challenge of task interference in complex learning scenarios. By integrating the Mixture-of-Experts (MoE) with LoRA, we present LoRA-MoE decoder, which delivers specialized learning paths for different tasks and modalities. After the validation across multiple modalities and tasks, Octavius alleviates the severe tug-of-war issue and achieves a significant performance boost in both 2D and 3D tasks.

**Limitations.** Compared to separate training on a single modality, introducing simultaneously multiple modalities for joint training may result in performance degradation, posing a challenge for future research. The combination of MLLMs and MoE still has great potential in addressing this problem, especially for a more complicated real-world system like embodied AI scenarios that require more modalities as input. Besides, we will further explore the token-based gate with load balancing strategies, especially when the number of downstream tasks increases.

**Acknowledgement.** This work is supported in part by National Key Research and Development Program of China (NO. 2021YFB1714300) , and National Natural Science Foundation of China (62132001).

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

## A   ADDITIONAL IMPLEMENTATION DETAILS

**Pre-training a language- and image-aligned Point-Bert following ULIP-like Pipeline.** To improve the generalization performance of instance-level point cloud encoder, we select ScanNet (Dai et al., 2017) as our dataset due to its diverse object categories instead of the original ULIP dataset. Besides, we devised a memory bank in our pre-training framework for fast convergence and better representative capabilities.

For specific, given the instance-level point cloud $\boldsymbol{P}_i \in \mathbb{R}^{N \times 6}$ within each candidate RoIs $\boldsymbol{r}_i$, we first retrieve images $\boldsymbol{I}_i$ from related regions using its camera intrinsic matrix, and generate a simple prompt template $\boldsymbol{L}_i$, *e.g.*, "`a photo of {CLASS}`", obtaining a multimodal triplet $\langle \boldsymbol{P}_i, \boldsymbol{I}_i, \boldsymbol{L}_i \rangle$. We then extract corresponding features with a pre-trained CLIP (Gao et al., 2023) and a trainable Point-Bert encoder (Yu et al., 2022):

$$\boldsymbol{h}_i^{\text{pcl}} = f^{\text{Point-Bert}}(\boldsymbol{P}_i); \ \boldsymbol{h}_i^{\text{img}} = f^{\text{CLIP}}(\boldsymbol{I}_i); \ \boldsymbol{h}_i^{\text{lang}} = f^{\text{CLIP}}(\boldsymbol{L}_i), \tag{8}$$

We then contrast $\boldsymbol{h}_i^{\text{pcl}}$ with $\boldsymbol{h}_i^{\text{img}}$ and $\boldsymbol{h}_i^{\text{lang}}$, bringing 3D representation closer to semantic information of images and language:

$$\mathcal{L}_{\text{contrast}} = w_1 \mathcal{L}_{\langle \text{pcl,img} \rangle} + w_2 \mathcal{L}_{\langle \text{pcl,lang} \rangle}, \tag{9}$$

where $\mathcal{L}_{\langle \text{pcl,img} \rangle}, \mathcal{L}_{\langle \text{pcl,lang} \rangle}$ are respective contrastive loss and $w_1, w_2$ are corresponding loss weights. As mentioned before, the memory bank $M$ is introduced to accommodate more negative samples for better feature alignment. For example, given $i$-th associated multimodal feature pair $\left\langle \boldsymbol{h}_i^{\text{pcl}}, \boldsymbol{h}_i^{\text{lang}} \right\rangle$, contrastive loss is given by

$$\mathcal{L}_{\langle \text{pcl,lang} \rangle} = -\sum_i \log \frac{\exp(\boldsymbol{h}_i^{\text{pcl}} \cdot \boldsymbol{h}_i^{\text{lang}} / \tau)}{\sum_{j \in \{i\} \bigcup M} \exp(\boldsymbol{h}_i^{\text{pcl}} \cdot \boldsymbol{h}_j^{\text{lang}} / \tau)}. \tag{10}$$

Eventually, the pre-trained Point-Bert is used for extracting instance-level 3D visual features that align with language and image in Object-As-Scene.

**LLM Architecture and Training Scheme.** We choose Vicuna-13B (Chiang et al.) as our LLM. Instructions are tokenized by SentencePiece (Kudo & Richardson, 2018). We apply LoRA-MoE on the language model for efficient fine-tuning and task-specific learning in all three setups. The number of experts in the above three setups is 4, 3, and 6, respectively. The rank of each LoRA expert is set to 32. During fine-tuning, we use an Adam (Kingma & Ba, 2014) optimizer with a total batch size of 64, a learning rate of $5 \times 10^{-4}$, and an epoch of 4 on all setups. All experiments are conducted on 4 NVIDIA A100 80GB GPUs.

Input images are all resized to 224×224 and split into 256 feature patches using CLIP ViT-L/14 (Radford et al., 2021). For point cloud data, we sample 1024 points from each RoI extracted by FCAF3D (Rukhovich et al., 2022) and generate corresponding features by pre-trained Point-Bert (Yu et al., 2022). Then we select $N_{\text{RoI}}$ instances with bbox confidence larger than a threshold $\tau = 0.3$ for each scene. Next, we use 16 queries in the fusion module to obtain aligned 3D visual features. Furthermore, in the multimodal setup, we pad the output 3D visual features to 256 with masks for aligning with image patches.

## B   ADDITIONAL EXPERIMENTS AND ABLATIONS

**The Tug-of-War Issues in Multimodal Learning.** We attempt to investigate the tug-of-war issues within the realm of multimodal learning. The results, delineated in Table 6, reveal that the tug-of-war issues not only prevail in multimodal learning, but also can be more severe. Here, we mainly focuses on 2D and 3D captioning tasks. When introducing more modalities during instruction tuning, a huge performance degradation can be observed, especially in 3D captioning tasks. After applying LoRA-MoE, the performance of 3D captioning tasks is enhanced, aligning with the levels achieved when fine-tuned on the single 3D modality. Meanwhile, the performance of 2D captioning is also greatly improved, underscoring the effectiveness of LoRA-MoE.

**Point Cloud Encoder.** As shown in Table 7, positional embedding (PE) improves the overall performance, since the position and scale of objects in PE can help the model better understand the

Table 6: We conduct another pilot study to reveal the tug-of-war issues in multimodal learning.

| FT. Dataset | MoE | Captioning | | | Avg. |
| --- | --- | --- | --- | --- | --- |
| | | Flickr30k (2D ZS.) | Scan2Cap (3D FT.) | NR3D (3D ZS.) | |
| LAMM v2 | | 0.21 | - | - | - |
| Scan2Inst | | - | 35.10 | 16.19 | - |
| LAMM v2+Scan2Inst | | 0.04 | 19.76 | 8.26 | - |
| LAMM v2+Scan2Inst | ✓ | 10.06 | 33.29 | 17.22 | 43.91% ↑ |

Table 7: **Ablation studies on point cloud encoder.** "PE" means positional embedding.

| #Queries | PE | Fused Modality | Cap. (Scan2Cap) | VQA (ScanQA) | Cls. (ScanNet) | Avg. |
| --- | --- | --- | --- | --- | --- | --- |
| | | | CIDEr | CIDEr | Acc | |
| 16 | Add | Lang. | 35.11 | 168.21 | 47.40 | 83.57 |
| 16 | Add | Lang. + Image | 45.00 | 160.33 | 61.60 | 88.97 |
| 64 | Add | Lang. | 41.45 | 161.69 | 48.80 | 83.98 |
| 256 | Add | Lang. | 19.36 | 164.55 | 48.40 | 77.44 |
| 16 | ✗ | Lang. | 29.39 | 168.98 | 42.60 | 80.32 |
| 16 | Concat | Lang. | 26.65 | 174.86 | 47.40 | 82.97 |

Table 8: **Additional ablation studies on MoE architecture.** "Ques." and "Sys." refer to using question or system prompt in the instruction as gate input, respectively.

| Gate Type | Gate Input | | Det. (VOC, IoU=0.5) | | VQA |
| --- | --- | --- | --- | --- | --- |
| | Ques. | Sys. | Recall | Prec. | Acc@1 |
| – (Baseline) | | | 7.61 | 5.95 | 40.31 |
| Sparse Top-2 | ✓ | | 39.04 | 35.21 | 46.95 |
| Sparse Top-1 | ✓ | | 22.42 | 21.23 | 36.88 |
| Sparse Top-3 | ✓ | | 38.57 | 36.02 | 43.89 |
| Sparse Top-2 | ✓ | ✓ | 34.23 | 30.78 | 40.25 |

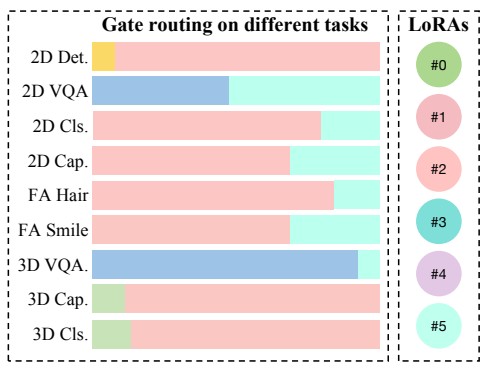

Figure 7: **Gate routing on 2D and 3D tasks.**

semantic information of the scene and instances. We select "Add" in our model due to its more balanced downstream results. We also ablate different numbers of learnable queries. Considering that we extract about 50 RoIs in the scene, if we use far more queries than this number, the overall performance will decrease. 16∼64 is a reasonable range for the number of queries. Furthermore, we attempt to enhance the semantic information by introducing image features corresponding to 3D RoIs using the cross-attention mechanism. Despite the significant improvement in captioning and classification tasks, the additional cost of rendering images based on point cloud limits its practical usage, making it only a supplementary method.

**Additional Ablations on LoRA-MoE.** The results are shown in Table 8. By incorporating the "System Prompt" as an adjunctive input alongside the question, a performance decline can be found possibly due to the redundant global information embedded within the system prompt (*e.g.*, "You are an AI assistant, ..."), which introduces ambiguity and hampers the efficacy of the gate prediction network.

Furthermore, we also explore the impact of using gates of different top-k selection. Specifically, the top-1 gate exhibits poor performance due to its limited flexibility in selection, offering only four combinations of different experts, in contrast to the more versatile top-2 and top-3 gate. In scenarios that employing the top-3 gate, we find that the contribution of the third choice is relatively small across most tasks. For instance, in classification tasks, the distribution of routing weights often resembles "[0.1, 0, 0.4, 0.5]", which is closely similar to the scores in top-2 gate. Consequently, the top-3 gate yields comparable performance achieved by the top-2 gate.

Table 9: **Ablation studies on OOD generalization of query in VQA evaluation.** "Ctx." and "Ques." denotes context and question, respectively. We report top 1 accuarcy on ScienceQA Lu et al. (2022) test dataset.

| Query Pattern | | VQA |
|---|---|---|
| Enriched Ctx. | Enriched Ques. | |
| | | 46.95 |
| ✓ | | 47.58 |
| | ✓ | 47.43 |
| ✓ | ✓ | 47.03 |

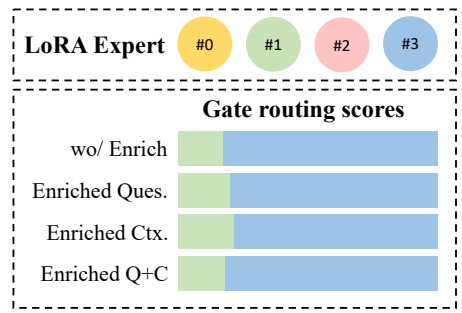

Figure 8: **Gate routing on different query pattern.** "Enriched Q+C" means using both enriched context and question as input.

**More Analysis and Discussion on Gate Routing.** As depicted in Figure 6, tasks such as classification, captioning and facial attribute recognition exhibit a tendency to engage similar experts with different routing scores. It can be attributed to underlying similarities among these tasks in training instructions and reponses, despite their varied downstream applications. For instance:

> • **Captioning**
>
> **Q:** What is the caption of the image?
>
> **A:** The image depicts an older man and a young man in a close embrace. The older man, who is bald and wearing a red shirt, appears to be hugging the younger man. They are standing in front of a black wall, which enhances the contrast of their positions.
>
> • **Classification**
>
> **Q:** What is the most accurate category label for this image?
>
> **A:** The image depicts a cute and adorable small white fluffy dog.
>
> • **Facial Attribute**
>
> **Q:** What color is the person's hair in the image?
>
> **A:** The image shows a young woman with dark, long, and curly hair.

These examples indicate that in tasks like classification and facial attribute recognition, LLMs tend to offer comprehensive descriptions of the target object (including attributes like color, shape, and descriptive adjectives) rather than mere categorical labels, which are very similar in captioning tasks. Therefore it is reasonable that the gating network makes similar expert selections for these tasks. Conversely, in very different tasks like detection, gating networks generate distinct expert selections.

Moreover, we jointly fine-tune Octavius on both LAMM v2 and Scan2Inst datasets, supplementing our analysis with an additional illustration of gate routing on 2D and 3D tasks, as presented in Figure 7. Similar to the observation in Figure 6, load balancing issues still occur within the distribution of gating scores, notably with experts #2 and #5. For the 3D modality, 3D captioning and 3D classification tasks mainly focus on instance-level perception, such as the caption or category of a specific object, while 3D VQA focuses more on inter-relations among multiple objects in the scene and the understanding of the entire scene. This divergence leads to two different pattern of routing weights between 3D captioning/classification and 3D VQA in Figure 7. Additionally, another interesting observation is the emergence of knowledge sharing across different modalities by certain experts (*e.g.*, expert #2), while others perfer for specilized modality (*e.g.*, experts #1 and #5).

**Generalizability of Instance-based Gate Routing.** To assess the generalizability of the proposed instance-based gate, we conduct several comprehensive ablation studies on the ScienceQA dataset Lu et al. (2022). The queries in ScicenceQA datasets comprise of distinct problem state-

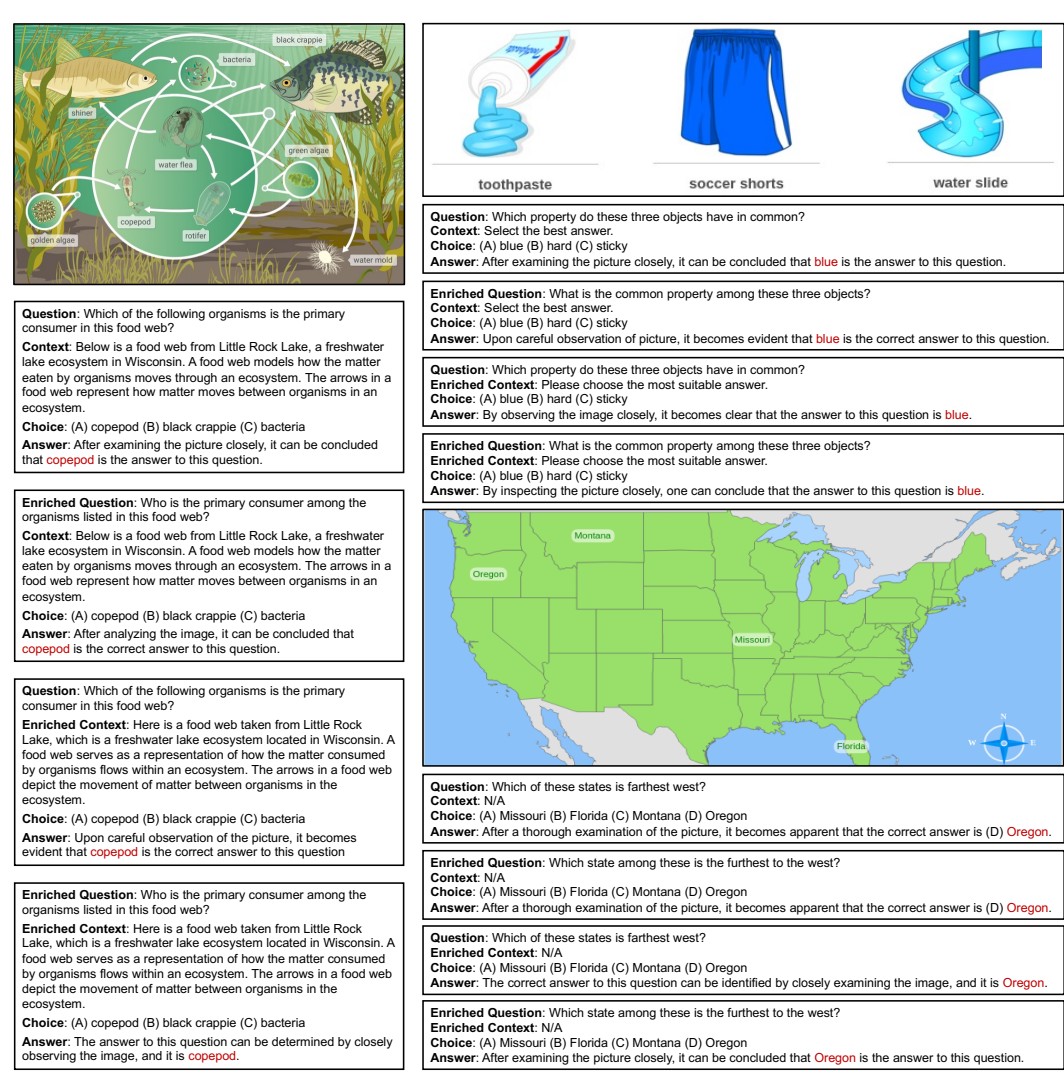

Figure 9: The response of Octavius given different query pattern in downstream VQA evaluation.

ments as well as contextual information. We employ GPT-3.5-turbo Brown et al. (2020) to enrich both the questions and the contexts separately, and ensure the enriched contents maintain consistency with the original semantics. Subsequently, we validate the proposed instance-based gate using these enriched questions and contexts. As detailed in Table 9, Octavius achieve stable performance across all enriched data, highlighting the strong generalization capacity of instance-based gate in processing input queries of different patterns. We also present several examples in Figure 9. Furthermore, we provide a comparative analysis of the routing weights between the default and enriched queries in Figure 8. Remarkably, the model consistently selects similar gates with comparable weights, regardless of the modifications in the data. This consistency demonstrates the robustness of Octavius in effectively managing VQA tasks, proficiently navigating questions and contexts of diverse structures and complexities.

**Complete Results on Downstream Tasks.** We provide complete experimental results for detection, captioning, and VQA tasks in all setups, as shown in Table 10, 11 and 12. We report recall and precision at IoU thresholds of 0.5 and 0.25 in detection tasks, and "BLEU-1/2/3/4", "CIDEr", "METEOR" and "ROUGE-L" in both captioning and VQA tasks.

Table 10: **Complete results on 2D downstream tasks.** In the detection task, we also provide recall and precision of the predicted bounding box without categories.

| MoE | Detection (PASCAL VOC) | | | | | | | |
|---|---|---|---|---|---|---|---|---|
| | w/ cls (IoU=0.5) | | wo/ cls (IoU=0.5) | | w/ cls (IoU=0.25) | | wo/ cls (IoU=0.5) | |
| | Recall | Prec. | Recall | Prec. | Recall | Prec. | Recall | Prec. |
| | 7.61 | 5.95 | 10.1 | 7.91 | 20.96 | 16.41 | 27.14 | 21.24 |
| ✓ | **39.04** | **35.21** | **44.16** | **39.63** | **51.38** | **46.12** | **59.19** | **53.13** |

| MoE | Captioning (Flickr30K) | | | | | | |
|---|---|---|---|---|---|---|---|
| | BLEU-1 | BLEU-2 | BLEU-3 | BLEU-4 | CIDEr | METEOR | ROUGE-L |
| | 13.283 | 7.328 | 3.733 | 1.883 | 0.21 | 12.482 | 17.707 |
| ✓ | **26.705** | **15.163** | **8.296** | **4.566** | **5.66** | **16.979** | **26.849** |

Table 11: **Complete results on 3D downstream tasks.** Here, [†] indicates the results of Scan2Cap is evaluated on a custom test set regenerated by 3D-LLM, which is different from ours.

| Models | MoE | Captioning (Scan2Cap) | | | | | | |
|---|---|---|---|---|---|---|---|---|
| | | BLEU-1 | BLEU-2 | BLEU-3 | BLEU-4 | CIDEr | METEOR | ROUGE-L |
| 3D-LLM[†] (Flamingo) | | 36.10 | 24.50 | 18.70 | 15.60 | – | 17.60 | 35.80 |
| Ours | | 34.16 | 20.92 | 12.45 | 7.56 | **39.56** | 13.03 | **32.66** |
| Ours | ✓ | **35.93** | **21.66** | **12.79** | **7.75** | 39.38 | **13.34** | 32.36 |

| Models | MoE | VQA (ScanQA) | | | | | | |
|---|---|---|---|---|---|---|---|---|
| | | BLEU-1 | BLEU-2 | BLEU-3 | BLEU-4 | CIDEr | METEOR | ROUGE-L |
| 3D-LLM (Flamingo) | | 30.30 | 17.80 | 16.00 | 7.20 | 59.20 | 12.20 | 32.30 |
| Ours | | 43.07 | 32.69 | 25.17 | **19.26** | 162.14 | 21.44 | **45.08** |
| Ours | ✓ | **44.24** | **33.16** | **25.24** | 19.16 | **167.31** | 21.44 | 44.87 |

| Models | MoE | Captioning (Nr3d) | | | | | | |
|---|---|---|---|---|---|---|---|---|
| | | BLEU-1 | BLEU-2 | BLEU-3 | BLEU-4 | CIDEr | METEOR | ROUGE-L |
| Ours | | 20.02 | 8.95 | 3.63 | 1.66 | 16.19 | 9.71 | 20.45 |
| Ours | ✓ | **21.16** | **10.00** | **4.38** | **2.07** | **17.22** | **11.06** | **22.37** |

Table 12: **Complete results on multimodal learning (2D & 3D).**

| MoE | Detection (PASCAL VOC) | | | | | | | |
|---|---|---|---|---|---|---|---|---|
| | w/ cls (IoU=0.5) | | wo/ cls (IoU=0.5) | | w/ cls (IoU=0.25) | | wo/ cls (IoU=0.5) | |
| | Recall | Prec. | Recall | Prec. | Recall | Prec. | Recall | Prec. |
| | 2.64 | 1.61 | 3.62 | 2.2 | 8.15 | 4.95 | 11.28 | 6.86 |
| ✓ | **34.3** | **25.07** | **38.97** | **28.48** | **47.11** | **34.43** | **54.72** | **39.99** |

| MoE | Captioning (Flickr30K) | | | | | | |
|---|---|---|---|---|---|---|---|
| | BLEU-1 | BLEU-2 | BLEU-3 | BLEU-4 | CIDEr | METEOR | ROUGE-L |
| | 14.335 | 8.132 | 4.288 | 2.274 | 0.038 | **13.673** | 17.083 |
| ✓ | **22.545** | **11.014** | **5.286** | **2.64** | **10.064** | 11.6 | **27.148** |

| MoE | Captioning (Scan2Cap) | | | | | | |
|---|---|---|---|---|---|---|---|
| | BLEU-1 | BLEU-2 | BLEU-3 | BLEU-4 | CIDEr | METEOR | ROUGE-L |
| | 26.16 | 14.79 | 7.91 | 4.36 | **13.76** | 29.26 | 19.76 |
| ✓ | **36.62** | **21.91** | **12.56** | **7.29** | 13.30 | **31.69** | **33.29** |

| MoE | VQA (ScanQA) | | | | | | |
|---|---|---|---|---|---|---|---|
| | BLEU-1 | BLEU-2 | BLEU-3 | BLEU-4 | CIDEr | METEOR | ROUGE-L |
| | **45.63** | **35.11** | **27.30** | 21.01 | **22.73** | **46.56** | **182.00** |
| ✓ | 44.48 | 34.20 | 26.76 | **21.04** | 22.23 | 46.22 | 181.44 |

| MoE | Captioning (Nr3d) | | | | | | |
|---|---|---|---|---|---|---|---|
| | BLEU-1 | BLEU-2 | BLEU-3 | BLEU-4 | CIDEr | METEOR | ROUGE-L |
| | 13.60 | 6.34 | 2.72 | 1.20 | 10.72 | 20.77 | 8.26 |
| ✓ | **20.96** | **9.95** | **4.27** | **2.02** | **11.13** | **22.29** | **17.22** |

## C   ADDITIONAL VISUALIZATION

In this section, we provide several responses of Octavius in Figure 10, 11 and  12.

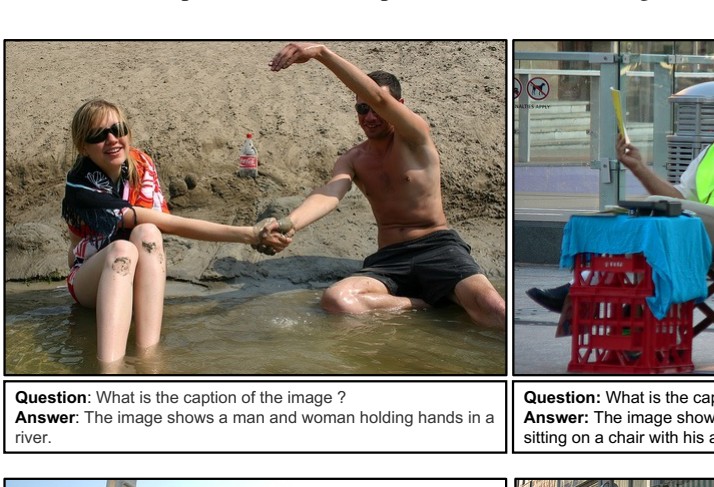

**Question**: What is the caption of the image ?
**Answer**: The image shows a man and woman holding hands in a river.

**Question:** What is the caption of the image ?
**Answer:** The image shows a man wearing a yellow safety vest, sitting on a chair with his arms extended.

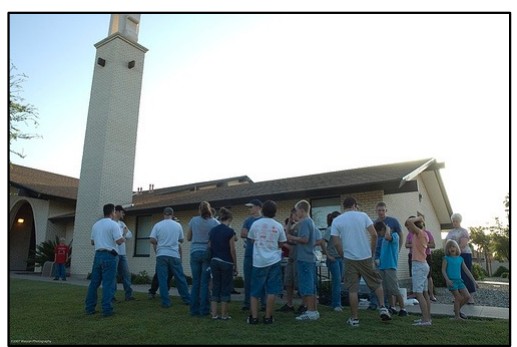

**Question**: What is the caption of the image ?
**Answer**: The image depicts a scene where a group of people are gathered near a tower with a clock face.

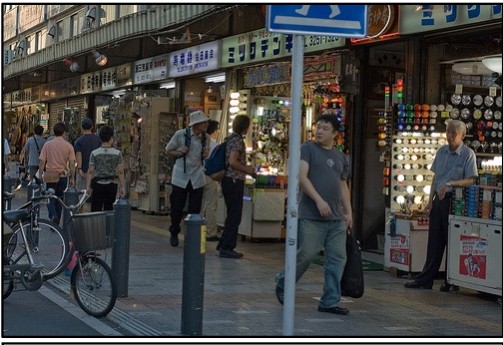

**Question**: What is the caption of the image ?
**Answer**: The image depicts a bustling street scene with numerous shops and vendors.

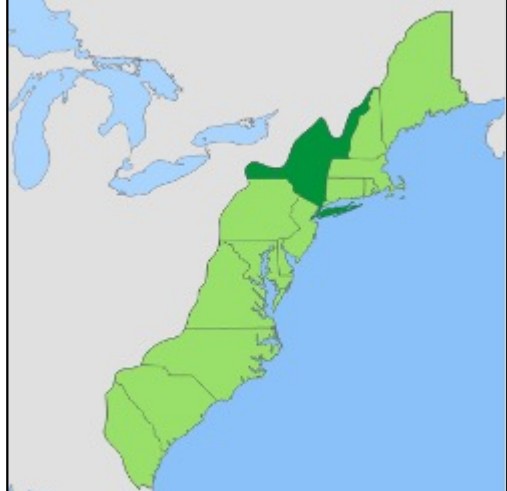

**Question**: What is the name of the colony shown?
**Context**: N/A
**Choice**: (A) New Hampshire (B) Vermont (C) New York (D) Delaware
**Answer**: Analysis of the image leads to the conclusion that New York is the correct response to this question.

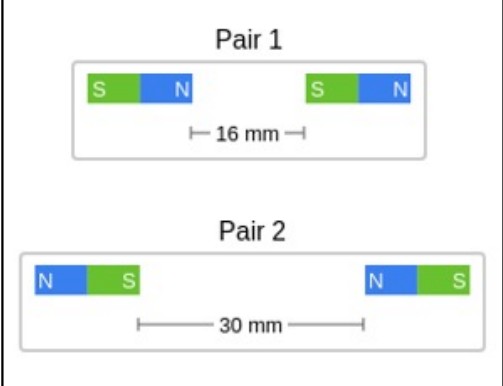

**Question**: Think about the magnetic force between the magnets in each pair. Which of the following statements is true?
**Context**: The images below show two pairs of magnets. The magnets in different pairs do not affect each other. All the magnets shown are made of the same material.
**Choice**: (A) The magnetic force is stronger in Pair 2. (B) The magnetic force is stronger in Pair 1. (C) The strength of the magnetic force is the same in both pairs.
**Answer**: By observing the image closely, it becomes clear that the answer to this question is B. The magnetic force is stronger in Pair 1.

Figure 10: The response of Octavius on 2D captioning and VQA.

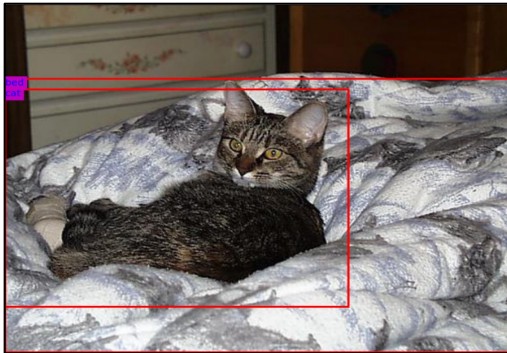

**Question**: Identify all objects in the image and provide their positions.
**Answer**: Positioned at [0.0, 0.24, 0.68, 0.86] within the image is an object that can be classified as belonging to the cat category, characterized by its unique texture and distinctive silhouette that differentiates it from other elements in the scene. This cat appears to be the focal point of the image, drawing the viewer's attention with its relaxed posture. At the [0.0, 0.21, 1.0, 0.99] position of the image, there is an object categorized as bed. The cat seems to be lying on the bed, adding a sense of tranquility and homeliness to the image.

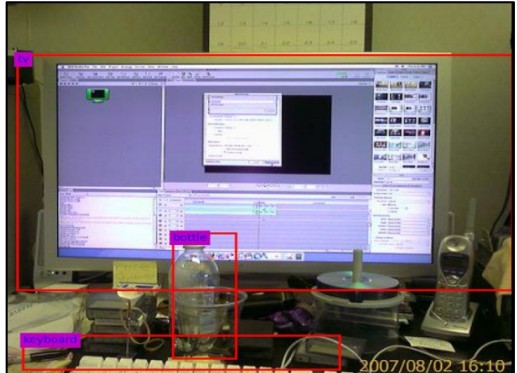

**Question**: Identify all objects in the image and provide their positions.
**Answer**: Positioned at [0.15, 0.89, 0.7, 0.98] within the image is an object that can be classified as belonging to the keyboard category, and this object is also the only one in the image that has a specific type of texture and a distinctive shape that sets it apart from the other objects. At the coordinates of [0.41, 0.62, 0.52, 0.95] position of the image, there exists an object categorized as bottle, characterized by its unique shape. Upon close inspection of the image, it can be observed that there is an object positioned at [0.14, 0.15, 1.0, 0.77] that belongs to the tv category.

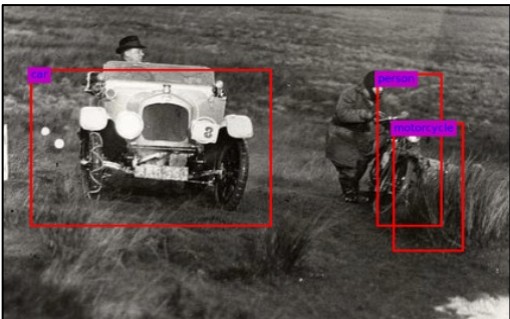

**Question**: Identify all objects in the image and provide their positions.
**Answer**: The object present at the [0.75, 0.42, 0.88, 0.73] coordinate in the image is classified as motorcycle. Positioned at [0.72, 0.3, 0.84, 0.67] within the image is an object that can be classified as belonging to the person category, and this object is also the only one in the image that has a specific type of texture and a distinctive shape that sets it apart from the other objects. At the coordinates of [0.07, 0.29, 0.52, 0.67] position of the image, there exists an object categorized as car.

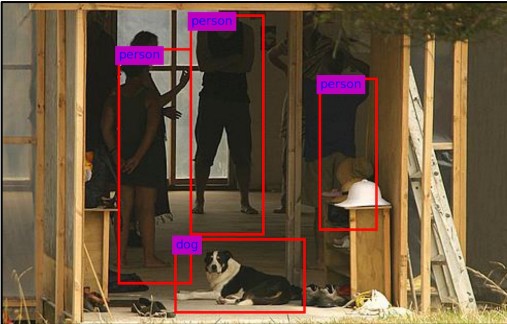

**Question**: Identify all objects in the image and provide their positions.
**Answer**: The [0.62, 0.39, 0.73, 0.7] position of the image contains an object that can be classified as person. At the [0.23, 0.33, 0.37, 0.81] position of the image, there is an object that can be categorized as person. The [0.37, 0.26, 0.51, 0.71] position of the image contains an object that can be identified as belonging to the person category. At the exact coordinates of [0.34, 0.72, 0.59, 0.87] in the image, there is an object that can be identified as belonging to the dog category.

Figure 11: The response of Octavius on 2D detection.

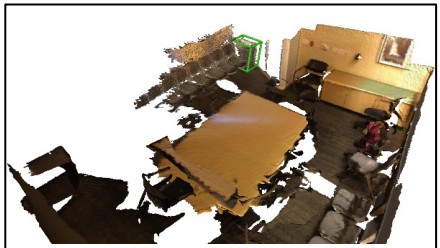

**Question**: What is the caption of the point cloud?
**Answer**: This is a trash can. it sets in the corner.

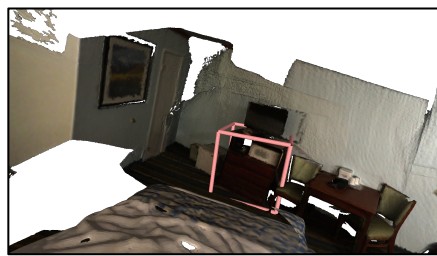

**Question**: What is the shape of the object that is in contact with a wall surface?
**Answer**: The object in the scene is rectangular in shape.

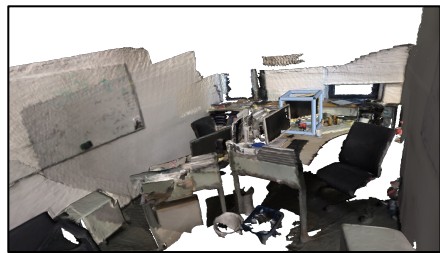

**Question**: Which side of the monitor is the shelf positioned on?
**Answer**: The shelf is on the right side of the monitor.

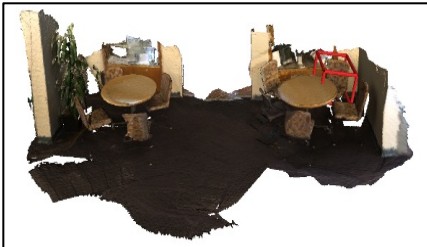

**Question**: What is located to the left of the display case?
**Answer**: A brown chair can be found in that position.

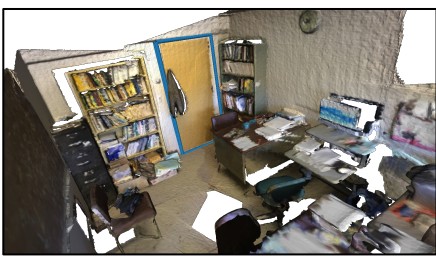

**Question**: What is the color of the door located in the corner?
**Answer**: The door situated in the corner is brown in color.

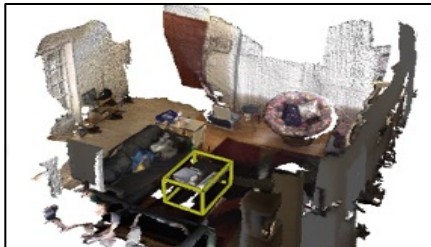

**Question**: What is the caption of the point cloud?
**Answer**: This is a black coffee table. it is in front of the couch.

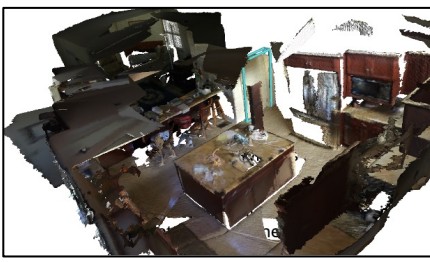

**Question**: What is the caption of the point cloud?
**Answer**: The door is northwest of the stove. The door is white and rectangular.

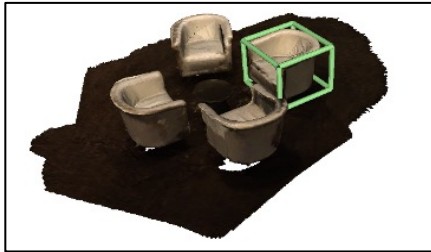

**Question**: What is the number of white chairs that encircle the brown end table?
**Answer**: There are four white chairs arranged in a circle around the brown table.

Figure 12: The response of Octavius on 3D captioning and VQA.

