# OpenReview forum: "Octavius: Mitigating Task Interference in MLLMs via LoRA-MoE"
_ICLR.cc/2024/Conference — ICLR 2024 poster_

### Official Review · Reviewer_7dhr · 2023-10-30

**Soundness:** 2 fair
**Presentation:** 2 fair
**Contribution:** 2 fair
**Rating:** 5
**Confidence:** 5

**Summary:**

The work proposes a multimodal LoRA-MoE decoder for task- and modality-specific learning. The experimental results (about 20% improvement) have shown the effectiveness and versatility of our design in various 2D and 3D downstream tasks.

**Strengths:**

1. The work adopts a simple but effective gate routing scheme allowing sparsely-activated LoRA modules to learn task- and modality-specific knowledge as an independent expert.
2. The work can address various 2D/3D vision and language tasks, and conduct various experiments to validate the effectiveness and versatility with a few trainable parameters.

**Weaknesses:**

1. The work focuses on SFT, whether it can be generalized to pre-training on massive data.
2. Lack of evaluation on some of the latest MLLM Chat evaluation, such as MMBench, MME, SEED-Bench, etc.
3. Lack of comparison with some classic and latest methods, such as MiniGPT4, mPLUG-Owl, LLAVA-1.5, Qianwen-VL, etc.

**Questions:**

1. In Figure 5, Loras2 has a probability of 0 on 6 tasks. What could be the possible reasons for this phenomenon? Could you provide more visualizations of experts and provide some insightful analysis?
2. Can you analyze the impact of k in the top-k sparse gate?

---

> ### Author Response · Authors · 2023-11-16
>
> > ***1. The work focuses on SFT, and whether it can be generalized to pre-training on massive data?***
>
>   - Generalizing to pre-training on massive data is not our focus. Our main motivation involves adding PEFT techniques (such as our LoRA-MoE) to the existing MLLMs to achieve better downstream performance generalization on more tasks and more modalities.
>
>   - Considering that other 2D MLLMs methods (except KOSMOS series, but they face replication problems) have also not mentioned training on massive datasets, for a fair comparison, we will not try to address this issue recently.
>
>   - Besides, collecting massive datasets for some modalities, such as point clouds, is quite challenging, thus preventing us from pre-training MLLMs on massive data in point cloud modality.
>
> > ***2. Lack of comparison with some classic and latest methods, such as MiniGPT4, mPLUG-Owl, LLAVA-1.5, Qianwen-VL, etc.***
>
>   - Our focus is not on proposing an entirely new SOTA 2D MLLM, but rather on addressing potential tug-of-war issues in downstream tasks by parameter-efficient fine-tuning existing MLLMs (such as LoRA). Therefore, direct comparison with the latest 2D MLLMs does not align with our motivation.
>
>   - For instance, as mentioned by Reviewer 3GST, InstructBLIP outperforms our baseline + LoRA-MoE on captioning/VQA tasks. It can only indicate that the baseline method used by InstructBLIP (BLIP-2) is much stronger than our chosen baseline method (LLaVA and LAMM). Additionally, such comparisons are completely unrelated to the motivation of our paper, because they cannot reflect improvements in task interference that our LoRA-MoE brings to the baseline model. Comparing the improvements with the baseline method itself on multi-task and multi-modal setup is more important, rather than comparing with other MLLMs. Therefore, we only compare the baseline model with LoRA and LoRA-MoE to explore the effectiveness of LoRA-MoE in this paper.
>
>   - A further discussion on this issue is why not use stronger baseline methods like InstructBLIP, KOSMOS series, and Shikra? It is because only LLaVA and LAMM are feasible options for us to adopt LoRA-MoE among existing 2D MLLM methods at that time.
>
>     - LLaVA-1.5 was released as a preprint on 10/5, exceeding the submission deadline of ICLR.
>
>     - KOSMOS series, InstructBLIP, and Qianwen-VL do not release their entire training datasets, so they face replication problems.
>
>     - Shikra mainly focuses on 2D grounding tasks. Although Shikra is capable of performing other tasks such as captioning and VQA, it does not conduct a zero-shot evaluation on captioning/VQA tasks (Flicker30k and VQAv2). If we directly remove related datasets (Flickr30k, VQAv2) from Shikra's training set for zero-shot evaluation, we cannot guarantee the zero-shot performance of Shikra on captioning/VQA tasks.
>
>     - Other 2D MLLM methods (like MiniGPT4, mPLUG-owl, etc.) do not significantly outperform LLaVA and LAMM. Considering the simplicity of LLaVA and LAMM, and both of them provide implementation with LoRA, we choose these two methods as our baseline.
>
>   - Besides, even if we ignore the motivation discussed above, all of the mentioned methods, including but not limited to MiniGPT4, mPLUG-Owl, LLAVA-1.5, Qianwen-VL, are trained on different datasets, different training strategies (e.g., PEFT vs. full-parameters FT, one-stage vs. two-stage), different pre-training weights (e.g., Vicuna-13b vs. Vicuna-7b, Vicuna vs. LLaMA vs. BLIP-2). Especially the differences  in datasets have a significant impact on the performance of MLLMs. Considering that our focus is not on dataset design but on model design, directly comparing their performance is unfair.
>
>   - Although such comparisons are unfair and do not align with our motivation, we still provide comparisons with some latest methods in 2D modality in Table E. An apparent example of unfairness is that many models in the table fail in the detection evaluation. Although their training sets contain COCO, most of them mainly focus on captioning tasks, and the localization capability is not sufficiently trained, so they achieve poor results on VOC. (Note that **Table A** is in the next comment.)

---

> ### Author Response · Authors · 2023-11-16
>
> **Table A**
>
> | Methods | FT. dataset | Training Strategy | VOC Recall@ 0.5 | ScienceQA Acc | Flickr30k CIDER | CIFAR10 Acc | Facial Hair Acc | Facial Smile Acc |
> | --- | --- | --- | --- | --- | --- | --- | --- | --- |
> | LLaMA-Adapter-v2 | S1: LAION400M/COYO/MMC4/SBU/Conceptual Captions/COCO S2: GPT4-LLM/LLaVA-150k | S1: prefix + projection S2: bias + norm | Failed  | 50.22  | 37.67  | 85.30  | 46.88 | 53.18 |
> | Otter  | MIMIC-IT 2.8M | 1.3B parameters | Failed | 38.67 | 68.75 | 81.22  | 56.40 | 56.60 |
> | Shikra  | S1: COCO/VG/Flickr30k/VQAv2 S2: LLaVA-150k/VCR/Shikra-RD | Whole LLM | 38.87  | 40.31 | 73.90 (non zero-shot evaluation) | 81.36  | 55.71 | 70.20 |
> | miniGPT4  | S1: CC/SBU/LAION S2: CC  | projection | Failed  | 43.43 | 5.68 | 62.49 | 43.47 | 66.36 |
> | mPLUG-owl   | S1: LAION/COYO/CC/COCO S2: Aplaca/Vicuna/Baize/LLaVA-150k | S1: visual encoder + abstractor S2: LoRA | Failed  | 36.39 | 26.15 | 82.32 | 40.93 | 51.32 |
> | LLaVA-LoRA (Vicuna-13b) | LLaVA-80k  | LoRA + projector | Failed  | 52.35 | 30.75 | 2.89 | 12.50 | 50.23 |
> | LLaVA-MoE | LLaVA-80k  | LoRA-MoE + projector | Failed | 55.58 | 23.08 | 41.00 | 3.93 | 52.17 |
> | LAMM | COCO/Bamboo/Locount/TextVQA | LoRA + projector | 6.68  | 40.85 | 0.09 | 73.40 | 43.26 | 53.37 |
> | LAMM-MoE | COCO/Bamboo/Locount/TextVQA | COCO/Bamboo/Locount/TextVQA | 39.04 | 46.95 | 5.66 | 65.40 | 60.93 | 59.82 |

---

> ### Author Response · Authors · 2023-11-16
>
> >***3. Lack of evaluation on some of the latest MLLM Chat evaluations, such as MMBench, MME, SEED-Bench, etc.***
>   - Existing MLLM evaluation frameworks receive many concerns regarding data quality and evaluation accuracy. They require long-term observation:
>     - Some MLLM evaluation frameworks like MMBench/MME construct their evaluation data by mixing the ground truth (GT) with confusing options, turning visual tasks into some kind of multi-choice questions. However, the quality of evaluation data generated by the automatic program or GPT requires careful examination. For example, (see this [link](https://openreview.net/forum?id=BfMQIJ0nLc&noteId=Nnj067ngbg), Reviewer g871) a lot of VQA samples in MMBench can be solved without looking at the images. In this situation, the visual perception capabilities of the MLLM are not well tested.
>     - Other MLLM evaluation frameworks like SEED-Bench evaluate the performance of MLLM through PPL [1]. However, SEED-Bench also faces some concerns regarding the evaluation accuracy (see #[7](https://github.com/AILab-CVC/SEED-Bench/issues/7)).
>     - We will continuously follow the development of MLLM evaluation frameworks, and conduct a comprehensive assessment when they are more mature. In this situation, we believe it is reasonable and effective to use traditional metrics, like Recall/Precision/CIDEr, to evaluate the performance of visual tasks.
>   - Based on the above discussion, we provide the results on MME (excluding MMbench on data quality concern and SEED-bench on evaluation accuracy concern) in **Table B**.
>
>     **Table B**
>     | methods | existence | count | position | color | posters | celebrity | scene | landmark | artwork | OCR | commonsense reasoning | numerical calculation | text translation | code reasoning | overall |
>     | --- | --- | --- | --- | --- | --- | --- | --- | --- | --- | --- | --- | --- | --- | --- | --- |
>     | LLaVA | 50.00 | 48.33 | 50.00 | 50.00 | 48.64 | 46.47 | 49.75 | 48.49 | 46.75 | 50.00 | 44.29 | 47.50 | 52.50 | 50.00 | 48.10 |
>     | LLaVA w/ MoE | 51.67 | 48.33 | 50.00 | 58.33 | 49.66 | 46.47 | 50.50 | 48.24 | 46.74 | 62.50 | 55.00 | 55.00 | 50.00 | 50.00 | 51.60 |
>
> [1] OpenNMT: Opensource toolkit for neural machine translation. 2017, Klein et al.
>
> >***4. In Figure 5, Loras2 has a probability of 0 on 6 tasks. What could be the possible reasons for this phenomenon?***
>   - Although LoRA#2 is not used in all 2D tasks (Figure 2), it is essential in the learning. We conducted experiments using three experts, as shown in **Table C**, and the performance is worse. It indicates that #2 is used in the middle-early stages of learning, but after convergence, the gate does not select #2 due to load balancing issues. We will add this table to the final version.
>
>     **Table C**
>     | Gate | #Experts | VOC Recall@ 0.5 | VOC Prec@ 0.5 | ScienceQA Acc@1 |
>     | --- | --- | --- | --- | --- |
>     |  Top-2 | 1 | 7.61 | 5.95 | 40.31 |
>     |  Top-2 | 3 | 21.64 | 12.20 | 44.72 |
>     |  Top-2 | 4 | 39.04 | 35.21 | 46.95 |
>   - #2 is not used due to the load balancing issue. As mentioned in the Ablation, the sample-based gate is incompatible with the load balancing strategy. For example, it is more reasonable to assign detection samples to a LoRA expert proficient in localizing than the other experts for the purpose of load balancing. Additionally, the experimental results in Table 6 confirm this viewpoint.
> >****5. Could you provide more visualizations of experts and provide some insightful analysis?****
>   - We provide the results on the combined LAMMv2 + Scan2Inst [here](    https://github.com/iclrxoct/iclr2024xoct/blob/main/6dGg_3.png).
>     - In the 2D+3D setup, there are still load balancing issues (#2 and #5) in the routing distribution like the 2D setup.
>     - In the 3D tasks, 3D captioning and 3D classification mainly focus on instance-level tasks, such as the caption or category of a specific object (see Figure 1 in the main part of the paper and Figure 4 in the appendix), while 3D VQA focuses more on relationships between multiple objects in the scene and the understanding of the scene. Therefore, the routing weights have two different patterns on 3D Cap./3D Cls and 3D VQA.
>     - Some experts engage in knowledge sharing across different modalities (#2), while others may be more inclined towards a single modality (#1 and #5).

---

> > ### Author Response · Authors · 2023-11-16
> >
> > >***6. Can you analyze the impact of k in the top-k sparse gate?***
> >   - We provide results of adopting different top-k gates in **Table D**. We will add this table to the final version.
> >
> >     **Table D**
> >     | Gate | #Experts | VOC Recall@ 0.5 | VOC Prec@ 0.5 | ScienceQA Acc@1 |
> >     | --- | --- | --- | --- | --- |
> >     | - | 1 | 7.61 | 5.95 | 40.31 |
> >     | top-1 | 4 | 22.42 | 21.23 | 36.88 |
> >     | top-2 | 4 | 39.04 | 35.21 | 46.95 |
> >     | top-3 | 4 | 38.57 | 36.02 | 43.89 |
> >     - The top-1 gate has poor performance due to its less flexibility in choice (only 4 combinations) compared with the top-2 and top-3 gate. While in the top-3 gate scenario, we find that the third choice is relatively small in most tasks (e.g., in classification tasks, the routing weight is like [0.1, 0., 0.4, 0.5]), therefore it is very similar to the top-2 gate, and achieves similar results as the top-2 gate.

---

> > > ### Author Response · Authors · 2023-11-22
> > >
> > > We hope this message finds you well. We have noted the deadline for open discussion of ICLR 2024 is approaching, yet we have not yet received any feedback from you. In light of this, we sincerely wish to know if we can receive any updated comments regarding to our submission 1570, titled "Octavius: Mitigating Task Interference in MLLMs via MoE". We are very pleased to hear from you on the reviewer’s comments.

---

### Official Review · Reviewer_3GST · 2023-10-31

**Soundness:** 2 fair
**Presentation:** 3 good
**Contribution:** 3 good
**Rating:** 5
**Confidence:** 5

**Summary:**

Observing the tug-of-war problem between 2D detection and VQA, this paper proposes a Mixture-of-Expert (MoE) style architecture with LoRA for efficient optimization of multi-modality and multi-task. To further support 3D instruction-following tasks, the authors propose a novel point cloud encoding architecture called Object-as-scene. The simple yet effective architecture demonstrate strong experimental performance.

**Strengths:**

- This paper constructs Octavius by collecting better detection annotation, combing multiple modality data and propose a novel MoE architecture.
- The authors provide a throughout discussion between related works.
- The proposed method is relatively simple but with strong performance.

**Weaknesses:**

- About the tug-of-war problem:
  - The authors demonstrate the existence of this problem simply by optimization 2D detection and VQA simultaneously.
  - Considering the authors also include multi-modality, it would be better to include preliminary also in this direction (e.g., 2D & 3D captioning).
- About the router input:
  - As shown in Fig. 2, the input consists of the system prompt, modality embedding and the question embedding, which contradicts wit Tab. 5.  Would you mind clarifying how exactly to construct the router input?
  - Also how to get the question embedding? Do you utilize other embedding model or just do that on-the-fly?

- About the MoE deployment:
  - Do you adopt LoRA-MoE in each Transformer block?
  - Do you adopt LoRA only for the MLP, similarly with the original MoE?
- About zero-shot evaluation:
  - According to Sec. 4.1, the zero-shot evaluation conducted in this paper is only about zero-shot evaluation on novel datasets of the training tasks (e.g., ScienceQA for evaluation and VQA for training).
  - Therefore, I wonder how it works if you have not seen any QA tasks during training, since it is difficult to understand a specialized architecture like MoE can generalize well to totally unseen tasks during pre-training.
  - Moreover, if you still train with VQA, but change the prompt template of ScienceQA during testing, will the MoE router be robust to this kind of OoD generalization?
- Overall, I think this is an interesting paper, but still with some problems to convince me about the effectiveness of the proposed method. I would consider increasing the score if my questions are well addressed.

**Questions:**

- Implementation details:
  - In Fig. 5, do you select a single layer to do the visualization or take an average of all layers?
  - In Tab. 6, the authors claim that loading balancing loss has negative effect. Do you use loading balancing afterwards? If not, how would we prevent the MoE architecture from collapsing to always using a single expert?
  - The 2D results in Tab. 4 have a significant performance gap with state-of-the-art VLLM like InstructBLIP (e.g., ScienceQA and Flickr30K captioning).
  - The authors should have utilized stronger baseline methods. There is a significant performance gap between LAMN and state-of-the-art methods including InstructBLIP and Shikra.

---

> ### Author Response · Authors · 2023-11-16
>
> > ***1. About the tug-of-war problem: The authors demonstrate the existence of this problem simply by optimizing 2D detection and VQA simultaneously. Considering the authors also include multi-modality, it would be better to include preliminary also in this direction (e.g., 2D & 3D captioning).***
>
> * First, in the pilot study, we demonstrate the existence of the tug-of-war on the whole LAMM-v2 datasets, including detection, VQA, caption, and classification tasks, not only optimizing detection and VQA.
>
> * We reorganize the related results of multimodal tug-of-war issues (in Table 4) in **Table A**.
>
>   **Table A**
>   | FT. dataset | MoE | 2D Caption (ZS. Flickr30k) | 3D Caption (FT. Scan) | 3D Caption (ZS. NR3D) | Avg. |
>   | --- | --- | --- | --- | --- | --- |
>   | 2D | X | 0.21 | - | - | - |
>   | 3D | X | - | 35.1 | 16.19 | - |
>   | 2D + 3D | X | 0.04 | 19.76 | 8.26 | - |
>   | 2D + 3D | O | 10.06 | 33.29 | 17.22 | 43.91% ⬆️ |
>
>   * From this table, we can find that the tug-of-war issues also exist in multimodal setup, and can even be more severe. Here we primarily compare 2D and 3D captioning tasks. When we introduce more modalities into instruction tuning, we can observe a huge performance drop, especially in 3D captioning tasks (35 -> 19, 16 -> 8). After applying LoRA-MoE, we can find that the performance of 3D captioning tasks can reach the level of performance that fine-tuned on the single 3D modality. Meanwhile, the performance of 2D captioning is also greatly improved.
>
> * This is a good perspective for analyzing the tug-of-war problems in the multimodal setup. We will add the tables and analysis to the final version. Thanks for your advice.
>
> > ***2. About the router input: As shown in Fig. 2, the input consists of the system prompt, modality embedding, and question embedding, which contradicts with Tab. 5. Would you mind clarifying how exactly to construct the router input? Also how to get the question embedding? Do you utilize other embedding model or just do that on-the-fly?***
>
> * We construct the router input following Equation 3, i.e., use question embedding only. And we obtain the question embedding on-the-fly.
>
> * We admit that Figure 2 is a little confusing. [Here](https://github.com/iclrxoct/iclr2024xoct/blob/main/3GST_1.png) is the updated version. We will modify it in the final version. Thanks for your advice.
>
> > ***3. About the MoE deployment: Do you adopt LoRA-MoE in each Transformer block? Do you adopt LoRA only for the MLP, similarly with the original MoE?***
>
> * In LoRA-MoE, LoRA experts are adopted in each Transformer block and each q/k/v/output projection layer. Specifically, we implement LoRA-MoE based on the original PEFT repository, so we replace all LoRA modules in LLMs with LoRA-MoE.
>
> * We will add these implementation details in the final version. Thanks for your advice.
>
> > ***4. In Fig. 5, do you select a single layer to do the visualization or take an average of all layers?***
>
> Octavius applies a global sample-based gating network, i.e., we employ one gating network to route LoRA experts in all layers, and Figure 5 illustrates overall gating scores.

---

> ### Author Response · Authors · 2023-11-16
>
> > ***5. About zero-shot evaluation: According to Sec. 4.1, the zero-shot evaluation conducted in this paper is only about zero-shot evaluation on novel datasets of the training tasks (e.g., ScienceQA for evaluation and VQA for training). Therefore, I wonder how it works if you have not seen any QA tasks during training, since it is difficult to understand a specialized architecture like MoE can generalize well to totally unseen tasks during pre-training. Moreover, if you still train with VQA, but change the prompt template of ScienceQA during testing, will the MoE router be robust to this kind of OoD generalization?***
>
> **Table B**
>
> | FT. dataset | MoE | ScienceQA | Caption | Det Recall50 | Cls |
> | --- | --- | --- | --- | --- | --- |
> | LAMM-v2 | X | 40.31 | 0.21 | 7.61 | 73.50 |
> | LAMM-v2 | O | 46.95 | 5.66 | 39.04 | 65.40 |
> | LAMM-v2 wo/ QA | X  | 14.97  | 0 | 20.64  | 72.06 |
> | LAMM-v2 wo/ QA | O | 12.59 | 0  | 36.55 | 67.61 |
>
> * We provide results in **Table B**. Specifically, we remove QA dialogues from the instruction dataset (including TextQA and common sense QA generated on COCO, reducing about 100k data in total) and test its downstream QA capability on ScienceQA datasets.
>
>   * From Table C, we find both the baseline model and MoE have significant performance drops in the ScienceQA evaluation, indicating MoE can not generalize well to unseen ScienceQA tasks during pre-training. Although generalizing to unseen tasks is not our primary focus, it is an interesting topic that we will research in the future.
>
>   * Considering the reduction in data amount (from 293k to 193k), there is a performance decrease in most downstream tasks. However, an interesting observation is that in the absence of MoE, the baseline, which is fine-tuned on the instruction dataset without QA datasets, achieves better results on detection tasks, proving the conflict between detection and QA tasks once again.
>
> * For OOD generalization in QA tasks, we provide experimental results in **Table C**. The query prompt of ScienceQA datasets contains three parts: question, context, and image. We enriched the question and context using GPT-3.5-turbo, respectively. We evaluate the ScienceQA Acc@1 using the 2D Octavius model.
>
>   **Table C**
>
>   | type | Default wo/ enrichment | Enriched contexts | Enriched questions | Enriched contexts + questions |
>   | --- | --- | --- | --- | --- |
>   | results | 46.95 | 47.58  | 47.43  | 47.03  |
>
>   * Our model shows improvements in both enriched contexts and questions, demonstrating strong OOD generalization and robustness of LoRA-MoE on varying questions and contexts. We also pick some examples [here](    https://github.com/iclrxoct/iclr2024xoct/blob/main/3GST_2.png).
>
>   * We also provide the comparison of the routing weights between the default template and the enriched template [here](      https://github.com/iclrxoct/iclr2024xoct/blob/main/3GST_3.PNG). **It is evident that regardless of the enriched parts, our model consistently selects similar gates with similar weights.** This observation serves as evidence for the robustness of our model in addressing VQA questions/contexts with different patterns.
>
> * We will add the tables, illustrations, and analyses to the final version.
>
> > ***6. In Tab. 6, the authors claim that loading balancing loss has negative effect. Do you use loading balancing afterwards? If not, how would we prevent the MoE architecture from collapsing to always using a single expert?***
>
> * We do not use a load balancing strategy afterwards. As mentioned in the section 4.3 (ablation), sample-based gate is incompatible with the load balancing strategy. For example, it is more reasonable to assign detection samples to a LoRA expert proficient in localizing than the other experts for the purpose of load balancing. Unless the dataset itself is balanced, it's difficult to equally assign experts. Additionally, the experimental results in Table 6 confirm this viewpoint. The load balancing strategy seems more suitable for token-based gates in conventional MoE models, but in this tug-of-war scenario, its interpretability is worse. Moreover, the experimental results of using token-based gates (also with load balancing strategy) are poor in Table 6.
>
> * We claim the load balancing issue as a limitation in the paper and will attempt to address it in future work.
>
> > ***7. The 2D results in Tab. 4 have a significant performance gap with state-of-the-art VLLM like InstructBLIP (e.g., ScienceQA and Flickr30K captioning).***
>
> InstructBLIP is built on powerful BLIP-2 (Vicuna-13b, caption 71.6 & ScienceQA 61.0, training with 129M caption/QA datasets), so it is reasonable that InstructBLIP has better results in caption and ScienceQA tasks.

---

> > ### Author Response · Authors · 2023-11-16
> >
> > > ***8. The authors should have utilized stronger baseline methods. There is a significant performance gap between LAMM and state-of-the-art methods including InstructBLIP and Shikra.***
> >
> > * InstructBLIP
> >
> >   * Compared with LLaVA/LAMM, InstructBLIP employs a more powerful 2D visual encoder (CLIP + BLIP-2 pretrained Q-former) while keeping the entire LLM frozen. Therefore, InstructBLIP and our LoRA-MoE are compatible and the LoRA/LoRA-MoE implementation of InstructionBLIP is feasible. For example, InstructBLIP freezes its LLM during instruction tuning, so we can add a third stage to fine-tune LLM with LoRA/LoRA-MoE while keeping other components frozen.
> >
> >   * However, although InstructBLIP mentions their templates to construct instruction data in the paper, a complete instruction-following dataset meta file has not been released. We cannot guarantee reproducing the results of InstructBLIP. You can find many issues in the InstructBLIP repository related to replication, e.g. [#302](https://github.com/salesforce/LAVIS/issues/302), [#439](https://github.com/salesforce/LAVIS/issues/439), [#542](https://github.com/salesforce/LAVIS/issues/542), and more. We will continuously track the feedback on the InstructBLIP Github repository, reproduce the results from its paper, and implement a LoRA/LoRA-MoE based InstructBLIP in the future.
> >
> > * Shikra
> >
> >   * Shikra applies a full fine-tuning scheme, we cannot guarantee the LoRA implementation of Shikra is effective.
> >
> >   * Besides, Shikra mainly focuses on 2D grounding tasks. Although Shikra is capable of performing other tasks such as captioning and VQA, it does not conduct a zero-shot evaluation on captioning/VQA tasks (test datasets, Flicker30k, and VQAv2, are included in the training set). If we directly remove related datasets (Flickr30k, VQAv2) from Shikra's training set for zero-shot evaluation, we cannot guarantee the zero-shot performance of Shikra on captioning/VQA tasks.
> >
> > * In addition to the mentioned InstructBLIP and Shikra, KOSMOS series [1-2] also faces replication difficulties (complete datasets + models not released + too excessive computational resources). Other 2D MLLM methods do not significantly outperform LLaVA and LAMM. Considering the simplicity of LLaVA and LAMM, and both of them implementing LoRA-based MLLM, we choose these two methods as our baseline. **Furthermore, we will continue to follow the developments in 2D MLLMs, and if a better baseline model emerges with sufficient transparency to support our work, we will adopt it.**
> >
> > [1] Language Is Not All You Need: Aligning Perception with Language Models. 2023, Huang et al.
> >
> > [2] KOSMOS-2: Grounding Multimodal Large Language Models to the World. 2023, Peng et al.

---

> > > ### Author Response · Authors · 2023-11-22
> > >
> > > We hope this message finds you well. We have noted the deadline for open discussion of ICLR 2024 is approaching, yet we have not yet received any feedback from you. In light of this, we sincerely wish to know if we can receive any updated comments regarding to our submission 1570, titled "Octavius: Mitigating Task Interference in MLLMs via MoE". We are very pleased to hear from you on the reviewer’s comments.

---

> > > > ### Comment · Reviewer_3GST · 2023-11-23
> > > > **Response to Author Rebuttal**
> > > >
> > > > Thanks for the detailed responses. My concerns mainly lie on two perspectives,
> > > >
> > > > **1. The choice of baseline methods.**
> > > >
> > > > Instead of answering why they do not select more suitable baseline methods, including InstructBLIP, Shirka and MiniGPT-4, the authors instead take Kosmos as a specific example to show LAMM is not worst baseline, ignoring the fact that all the aforementioned baseline methods are open to public at close time with LAMM. Considering the similarity of pre-training datasets among these baseline methods, it is highly possible that the proposed method cannot scale to better baselines.
> > > >
> > > > **2. The MoE architecture utilized in this paper.**
> > > >
> > > > I tend to agree with Reviewer 6dGg and 7dhr that there are many evidences suggesting that the reported results might be obtained via delicate designs instead of the flexible MoE achitecture, including 1) Octavius utilizes a global router for all the MoE layers instead of one for each layer, 2) activation maps in Figure 5 suggest that there are only two activate experts with one expert even totally not activated. It seems that the learnt model does not work as the authors claim in this paper.
> > > >
> > > > Therefore, it is hard to convince me the scalabiltiy of the proposed method for base models beyond LAMM and the considered 2D and 3D captioning and detection tasks. I choose to hold my initial judgment about this paper.

---

> > > > > ### Author Response · Authors · 2023-11-23
> > > > >
> > > > > > ***1. The choice of baseline methods.***
> > > > >
> > > > > - Actually, we have verified our design of LoRA-MoE not only on LAMM, but also on LLaVA.
> > > > > - "Instead of answering why they do not select more suitable baseline methods, including InstructBLIP, Shirka and MiniGPT-4, the authors instead take Kosmos as a specific example to show LAMM is not worst baseline"
> > > > >   - We have explained in the [rebuttal](https://openreview.net/forum?id=rTDyN8yajn&noteId=5ANzly7Y5E) why we do not select suitable baseline methods, including InstructBLIP, Shirka and MiniGPT-4. And we do not pick Kosmos as a specific example.
> > > > >     > - InstructBLIP
> > > > >     >   - Compared with LAMM/LLaVA, InstructBLIP employs a more powerful 2D visual encoder (CLIP + BLIP-2 pretrained Q-former) while keeping the entire LLM frozen. InstructBLIP and our LoRA-MoE are compatible and the LoRA/LoRA-MoE implementation of InstructionBLIP is feasible. For example, InstructBLIP freezes its LLM during instruction tuning, so we can add a third stage to fine-tune LLM with LoRA/LoRA-MoE while keeping other components frozen.
> > > > >     >   - However, although InstructBLIP mentions their templates to construct instruction data in the paper, a complete instruction-following dataset meta file has not been released. We cannot guarantee reproducing the results of InstructBLIP. You can find many issues in the InstructBLIP repository related to replication, e.g. #302, #439, #542, and more. We will continuously track the feedback on the InstructBLIP Github repository, reproduce the results from its paper, and implement a LoRA/LoRA-MoE based InstructBLIP in the future.
> > > > >     > - Shikra
> > > > >     >   - Shikra applies a full fine-tuning scheme, we cannot guarantee the LoRA implementation of Shikra is effective.
> > > > >     >   - Besides, Shikra mainly focuses on 2D grounding tasks. Although Shikra is capable of performing other tasks such as captioning and VQA, it does not conduct a zero-shot evaluation on captioning/VQA tasks (test dataset, Flicker30k, and VQAv2, are included in the training set). If we directly remove related datasets (Flickr30k, VQAv2) from Shikra's training set for zero-shot evaluation, we cannot guarantee the zero-shot performance of Shikra on captioning/VQA tasks.
> > > > >     > - In addition to the mentioned InstructBLIP and Shikra, KOSMOS series [1-2] also faces replication difficulties (complete datasets and models not released + too excessive computational resources). Other 2D MLLM methods do not significantly outperform LLaVA and LAMM. Considering the simplicity of LLaVA and LAMM, and both of them implementing LoRA-based MLLM, we choose these two methods as our baseline. Furthermore, we will continue to follow the developments in 2D MLLMs, and if a better baseline model emerges with sufficient transparency to support our work, we will adopt it.
> > > > >
> > > > > - "ignoring the fact that all the aforementioned baseline methods are open to public at close time with LAMM."
> > > > >   - Aforementioned baseline methods has primarily released their pretraining weights and demo, but their instruction-following datasets has not been open to public.
> > > > > - "Considering the similarity of pre-training datasets among these baseline methods"
> > > > >   - Although these training data seem similar, the quality and scale of different dataset can greatly affect the performance. For example, in issue [#439](https://github.com/salesforce/LAVIS/issues/439) (a re-implementation issue following the InstructBLIP paper), the replicated performance visual dialog tasks have dropped by 20% (from 46.6 in paper -> 38), which is significant.

---

> > > > > > ### Author Response · Authors · 2023-11-23
> > > > > >
> > > > > > > ***2. The MoE architecture utilized in this paper.***
> > > > > >
> > > > > > - "the reported results might be obtained via delicate designs instead of the flexible MoE architecture"
> > > > > >   - Our LoRA-MoE is straightforward, which is just  a mixture of LoRAs that are globally routed. We have not tried a delicate design that is specially designed for LAMM. Note that our method also works on LLaVA, which has quite a different language-based decoder than that of LAMM (i.e., LLaMA versus Vicuna).
> > > > > >
> > > > > > - "Octavius utilizes a global router for all the MoE layers instead of one for each layer"
> > > > > >   - In our case, using global router is a reasonable choice for most MLLMs, because by design the input question embedding for router should be consistent across every layer. This is also a simple and more interpretable design: choosing a specific expert for a type of task. The layer-wise gate seems more suitable for token-based gates.
> > > > > >   - Additionally, we also try to add the independent gate network to every layer. However, the results are a bit lower and unstable.
> > > > > >
> > > > > > - "activation maps in Figure 5 suggest that there are only two activate experts with one expert even totally not activated."
> > > > > >
> > > > > >   - Compared with using only 2 experts (either learnable or average), the performance improvement of LoRA-MoE is remarkable, which means **training more experts benefit the inference, even though only two of them are activated**.
> > > > > >
> > > > > >     > We provide the results of the model with 2 experts in Table A. In the "2-learnable" experiment, we apply normalized gating scores (a 2-dim tensor, e.g., [0.34, 0.66]) as expert weights, while in "2-average" experiments, we use [0.5, 0.5] as expert weights.
> > > > > >
> > > > > >     > **Table A**
> > > > > >     > | Gate Type | Gate Input | #Experts | LoRA-Rank  | Det Recall  | Det Prec.  | VQA Acc | # Trainable Params |
> > > > > >     > | --- | --- | --- | --- | --- | --- | --- | --- |
> > > > > >     > |  Baseline | - | 1 | 32 | 7.61 | 5.95 | 40.31 | 0.4% |
> > > > > >     > |       Top-2 | Question | 4 | 32 | 39.04 | 35.21 | 46.95 | 1.6% |
> > > > > >     > |  Top-2 | Question| 2-learnable | 32 | 15.03 | 8.63 | 41.40 | 0.8% |
> > > > > >     > |  Top-2 | Question | 2-average | 32 | 7.28 | 5.32 | 40.80 | 0.8% |
> > > > > >
> > > > > >     - In Figure 5, the routing combination mainly comprising two experts is due to the load balancing issue. The lower weights of other experts mean that these experts may not usually be activated, but do not mean that they are useless. As what we have discussed, experts indicate specific types of tasks or instructions. The low frequency of an expert in Figure 5 may indicate that their data is of low frequency, which is mentioned as the load balancing issue in our paper. We will provide more experiments to validate the usage of each expert in the future version.
> > > > > >
> > > > > > - "it is hard to convince me the scalabiltiy of the proposed method for base models beyond LAMM and the considered 2D and 3D captioning and detection tasks."
> > > > > >   - We also provide the results of LLaVA in Table 2. We will provide more comparisons when a better baseline model emerges with sufficient transparency to support our work in the future.

---

### Official Review · Reviewer_6dGg · 2023-11-01

**Soundness:** 2 fair
**Presentation:** 4 excellent
**Contribution:** 2 fair
**Rating:** 6
**Confidence:** 4

**Summary:**

This paper contributes a method for mitigating task interference in instruction tuning by learning LoRA-based Mixture-of-Experts. By using a sparse gate routing scheme, different LoRA experts can learn task- and modality-specific knowledge. In experiments, they do instruction tuning for MLLMs on both 2D image tasks and 3D point cloud tasks, each having individual vision encoders.

**Strengths:**

- The paper is well-motivated -- MoEs have been shown to be useful for distributing the different types of knowledge that are required for multi-task learning, and VL instruction tuning is a good application of this insight.

- The experiments are performed on both 2D image and 3D point cloud tasks, both individually and with the two datasets combined.

**Weaknesses:**

- I am primarily concerned by the analysis in Figure 5 -- it seems that all the 2D tasks are using only two experts! This makes me skeptical about the utility of MoE at all.  Could you run the ablation in Table 5 with 2 experts? so these two experts will get selected each time and there in routing involved, but the model has the capacity to learn with 2 LoRAs at once instead of one (which is what it seems to be doing in Fig5)

- An additional analysis that is needed is how the gate routing is distributed between 2D and 3D tasks, for the model that is trained on the combined LAMMv2+ScanNet instruction tuning dataset.

Given the result in Figure 5, I am not convinced about the utility of MoE with routing, when it doesn't seem that different experts are even used.

- I am not sure what the non-MoE baseline in Tables 2-3 is -- is it merely training the frozen model minus the LoRA parameters?

**Questions:**

- What is the actual LLM decoder that you use? it's never mentioned throughout the paper.

---

> ### Author Response · Authors · 2023-11-16
>
> > ***1. I am primarily concerned by the analysis in Figure 5 -- it seems that all the 2D tasks are using only two experts! This makes me skeptical about the utility of MoE at all. Could you run the ablation in Table 5 with 2 experts? so these two experts will get selected each time and there in routing involved, but the model has the capacity to learn with 2 LoRAs at once instead of one (which is what it seems to be doing in Fig5)***
>
> * We use top-2 gate for routing, i.e., selecting two experts from all experts for predicting. It is a very popular and common gating strategy used in MoE models [1-4].
>
> * We provide the results of the model with 2 experts in **Table A**. In the "2-learnable" experiment, we apply normalized gating scores (a 2-dim tensor, e.g., [0.34, 0.66]) as expert weights, while in "2-average" experiments, we use [0.5, 0.5] as expert weights.
>
>   **Table A**
>
>   | Gate Type| #Experts | LoRA-Rank  | VOC Recall@ 0.5  | VOC Prec@ 0.5  | ScienceQA Acc@1 | # Trainable Params. |
>   |:---:|:---:|:---:|:---:|:---:|:---:|:---:|
>   | Baseline | 1 | 32 | 7.61 | 5.95 | 40.31 | 0.4% |
>   | top-2 | 4 | 32 | 39.04 | 35.21 | 46.95 | 1.6% |
>   | top-2 | 2-learnable | 32 | 15.03 | 8.63 | 41.40 | 0.8% |
>   | top-2 | 2-average | 32 | 7.28 | 5.32 | 40.80 | 0.8% |
>
>   * We can observe a huge performance gap with 2 experts in detection (VOC) tasks. For the "2-average" experiment that directly applies [0.5, 0.5] as expert weights, it can be seen as simply increasing the model parameters. For example, you can replace 2 experts with a larger LoRA', i.e., LoRA' = 0.5*LoRA1 + 0.5*LoRA2. For the "2-learnable" experiment that applies weights generated by the gate, the gate can learn more kinds of combinations of routing weights for different tasks. To some extent, different combinations of routing weights can be seen as different routing results. Therefore, it achieves better results than the "2-average" experiment.
>
>   * When we use more experts (from 2 to 4), the gate has more choices and achieves the best results in **Table A**. However, as more experts are introduced, as mentioned in the section 4.3 (ablation), the model will be affected by the load balancing issues, leading to a decline in the utilization rate of different experts (this is why LoRA#2 is not used in Figure 5). Allocating too many tasks to the same experts eventually leads to a performance drop (8 experts perform lower than 4 experts in Table 5).
>
> * We will add the above results in Table 5 in the final version.
>
> [1] Outrageously Large Neural Networks: The Sparsely-Gated Mixture-of-Experts Layer. 2017, Shazeer et al.
>
> [2] GShard: Scaling Giant Models with Conditional Computation and Automatic Sharding. 2022, Lepikhin et al.
>
> [3] GLaM: Efficient Scaling of Language Models with Mixture-of-Experts. 2022, Du et al.
>
> [4] Lifelong Language Pretraining with Distribution-Specialized Experts. 2023, Chen et al.
>
> > ***2. An additional analysis that is needed is how the gate routing is distributed between 2D and 3D tasks, for the model that is trained on the combined LAMMv2+ScanNet instruction tuning dataset.***
>
> * We provide the visualization of routing results on the combined LAMMv2 + Scan2Inst [here](    https://github.com/iclrxoct/iclr2024xoct/blob/main/6dGg_3.png).
>
> * In the 2D+3D setup, there are still load balancing issues (#2 and #5) in the routing distribution like the 2D setup. (see the discussion of the load balancing issues in section 4.3, ablation)
>
> * In the 3D tasks, 3D captioning and 3D classification mainly focus on instance-level tasks, such as the caption or category of a specific object (see Figure 1 in the main part of the paper and Figure 4 in the appendix), while 3D VQA focuses more on relationships between multiple objects in the scene and the understanding of the entire scene. Therefore, there are two patterns of routing weights between 3D Cap./3D Cls and 3D VQA.
>
> * Some experts engage in knowledge sharing across different modalities (#2), while others may be more inclined towards a single modality (#1 and #5).

---

> > ### Author Response · Authors · 2023-11-16
> >
> > > ***3. Given the result in Figure 5, I am not convinced about the utility of MoE with routing, when it doesn't seem that different experts are even used. I am not sure what the non-MoE baseline in Tables 2-3 is -- is it merely training the frozen model minus the LoRA parameters?***
> >
> > * First, non-MoE baselines in Table 2-3 train additional LoRAs from scratch (one LoRA per layer) while keeping the pre-trained weights of Vicuna-13b frozen following LAMM [1]. This is a common PEFT training setup for LLMs.
> >
> > * In Figure 5, Classification/Caption/FacialAttribute share similar expert selection but have different weights. It is because, **although these tasks have different downstream applications, they are very similar in instructions and responses.** I can give some examples below:
> >
> >   * Captioning:
> >
> >     **Q**: What is the caption of the image?
> >
> >     **A**: The image depicts an older man and a young man in a close embrace. The older man, who is bald and wearing a red shirt, appears to be hugging the younger man. They are standing in front of a black wall, which enhances the contrast of their positions.
> >
> >   * Classification:
> >
> >     **Q**: What is the most accurate category label for this image?
> >
> >     **A**: The image depicts a cute and adorable small white fluffy dog.
> >
> >   * Facial Attribute
> >
> >     **Q**: What color is the person's hair in the image?
> >
> >     **A**: The image shows a young woman with dark, long, and curly hair.
> >
> >   From the above examples, we can find that in classification/facial attribute tasks, **LLMs are inclined to provide detailed descriptions of the target object (including color, shape, adjectives, etc.) rather than just basic categories**. The responses to these tasks are very similar to some simple captioning tasks, so it is reasonable that the gating network makes similar expert selections for captioning/classification/facial attribute tasks. Instead, in very different tasks like detection, gating networks generate different expert selections.
> >
> > [1] Lamm: Language-assisted multi-modal instruction-tuning dataset, framework, and benchmark. 2023, Yin et al.
> >
> > > ***4. What is the actual LLM decoder that you use? it's never mentioned throughout the paper.***
> >
> > We use Vicuna-13b following LAMM [1] and LLaVA [2] in experiments. It is mentioned in the supplementary.
> >
> > [1] Lamm: Language-assisted multi-modal instruction-tuning dataset, framework, and benchmark. 2023, Yin et al.
> >
> > [2] LLaVA: Visual Instruction Tuning. 2023, Liu et al.

---

> > > ### Author Response · Authors · 2023-11-22
> > >
> > > We hope this message finds you well. We have noted the deadline for open discussion of ICLR 2024 is approaching, yet we have not yet received any feedback from you. In light of this, we sincerely wish to know if we can receive any updated comments regarding to our submission 1570, titled "Octavius: Mitigating Task Interference in MLLMs via MoE". We are very pleased to hear from you on the reviewer’s comments.

---

### Official Review · Reviewer_WqWj · 2023-11-10

**Soundness:** 4 excellent
**Presentation:** 4 excellent
**Contribution:** 4 excellent
**Rating:** 8
**Confidence:** 3

**Summary:**

The paper mainly introduced 1) LoRA-MoE (Mixture-of-Experts) decoder to mitigate tug-of-war (interference) problem between different tasks/modalities, and 2) point cloud encoder called Object-As-Scene to extract language-aligned scene-level 3D features.
Experiments on both 2D and 3D tasks show LoRA-MoE improves performance by ~20% over strong baselines like LAMM and LLaVA-LoRA. The model is also verified on multimodal learning with both images and point clouds.

**Strengths:**

- The idea of MoE with sample routing to mitigate task interference for MLLMs is novel.
- The author conducted thorough experiments to validate the framework on a diverse set of 2D and 3D tasks. The gains are substantial.
- The framework is modular and extensible to incorporate more modalities and tasks.

**Weaknesses:**

- There is no analysis on how the routing among experts actually works. It would be great if the authors can provide some qualitative study of the predictions from sample-based gating network as responses to the input task, to show how the routing mechanism work. I wonder whether the gating network will simply act like a task classifier, or it's not the case.
- The scaling behavior as more modalities and tasks are added is not studied. There may be limitations in very high multi-task settings.

**Questions:**

- Can you show some qualitative study of the predictions from the sample-based gating network as responses to the input task? It can help us understand how network routing works.

---

> ### Author Response · Authors · 2023-11-16
>
> > ***1. There is no analysis on how the routing among experts actually works. It would be great if the authors can provide some qualitative study of the predictions from sample-based gating network as responses to the input task, to show how the routing mechanism work. I wonder whether the gating network will simply act like a task classifier, or it's not the case.***
>
> * We provide the visualization of routing weights on 2D setup in Figure 5.
>
>   * In most downstream tasks, the questions of each task are the same in evaluation. Therefore, Figure 5 can be seen as the predictions from the gating network as responses to the input tasks. The only exception is ScienceQA task where different questions lead to different routing weights. We analyze the generated weights for all questions in ScienceQA and find that 3 experts (#0/1/3) are chosen in ScienceQA, but most of them are composed of LoRA#1 and LoRA#3. Therefore, we use the normalized average weights of #1 and #3 for visualization.
>
>   * Furthermore, we find that captioning/classification/facial attribute tasks share similar expert selection in Figure 5. It is because, although these tasks have different downstream applications, they are very similar in instructions and responses. I can give some examples below:
>
>     * Captioning:
>
>       **Q**: What is the caption of the image?
>
>       **A**: The image depicts an older man and a young man in a close embrace. The older man, who is bald and wearing a red shirt, appears to be hugging the younger man. They are standing in front of a black wall, which enhances the contrast of their positions.
>
>     * Classification:
>
>       **Q**: What is the most accurate category label for this image?
>
>       **A**: The image depicts a cute and adorable small white fluffy dog.
>
>     * Facial Attribute
>
>       **Q**: What color is the person's hair in the image?
>
>       **A**: The image shows a young woman with dark, long, and curly hair.
>
>     From the above examples, we can find that in classification/facial attribute tasks, LLMs are inclined to provide detailed descriptions of the target object (including color, shape, adjectives, etc.) rather than just basic categories. They are similar to some simple captioning tasks, so it is reasonable that the gating network makes similar expert selections for captioning/classification/facial attribute tasks. **In these cases, the gating network can learn to allocate similar experts to similar but different tasks, and it can not be simply regarded as a task classifier. While in very different tasks like detection, the gating network tends to generate different expert selections.**
>
> * We also provide additional visualization on 2D+3D setup [here](https://github.com/iclrxoct/iclr2024xoct/blob/main/6dGg_3.png).
>
>   * In the 2D+3D setup, there are still load balancing issues (#2 and #5) in the routing distribution like the 2D setup. (see the discussion of the load balancing issues in section 4.3, ablation)
>
>   * In the 3D tasks, 3D captioning and 3D classification mainly focus on instance-level tasks, such as the caption or category of a specific object (see Figure 1 in the main part of the paper and Figure 4 in the appendix), while 3D VQA focuses more on relationships between multiple objects in the scene and the understanding of the entire scene. Therefore, there are two patterns of routing weights between 3D Cap./3D Cls and 3D VQA.
>
>   * Some experts engage in knowledge sharing across different modalities (#2), while others may be more inclined towards a single modality (#1 and #5).
>
> > ***2. The scaling behavior as more modalities and tasks are added is not studied. There may be limitations in very high multi-task settings.***
>
> * In the 2D/3D/2D+3D setups, our work already covers a wide range of tasks in most 2D/3D scenarios, including captioning, VQA, classification, etc., in both instruction tuning and downstream evaluations. Recent works such as VL-BERT[1], QWen-VL[2], and VisionLLM[3] involve 2D visual tasks that are similar to ours. If we want to further expand more datasets in these setups, it would primarily be expansions in terms of data amount.  Since this is a very high multitask learning in these setups, we believe that we can adequately demonstrate the generalization capability of LoRA-MoE in multitask learning and multimodal learning on the existing datasets and modalities.
>
> * Considering the workload, we will further explore more modalities on this topic beyond 3D like audio, and robotic tasks in future works.
>
> [1] Vl-beit: Generative vision-language pretraining. 2022, Bao et al.
>
> [2] Qwen-vl: A frontier large vision-language model with versatile abilities. 2023, Bai et al.
>
> [3] Visionllm: Large language model is also an open-ended decoder for vision-centric tasks. Wang et al.

---

> > ### Comment · Reviewer_WqWj · 2023-11-21
> > **Thanks for the response**
> >
> > The response addressed my concerns. I will keep my score unchanged.

---

> > > ### Author Response · Authors · 2023-11-22
> > >
> > > Thanks for your reply. We will take your advice to improve the paper.

---

### Official Review · Reviewer_vGo5 · 2023-11-10

**Soundness:** 3 good
**Presentation:** 3 good
**Contribution:** 3 good
**Rating:** 8
**Confidence:** 4

**Summary:**

This paper proposes to use a combination of MoE with the LoRA technique to address the challenge of incorporating additional tasks in the MLLM. The MoE components are chosen using a sparsely gating network. The outcomes demonstrate the efficacy of this approach.

**Strengths:**

This method is straightforward but powerful, capable of integrating numerous vision tasks within this framework. Its adaptability is showcased as it seamlessly operates with both 2D images and 3D point clouds. Ultimately, the integration of LoRA with MoE for PEFT proves to be highly efficient.

**Weaknesses:**

Given our awareness of each example's task, an important baseline involves employing a dedicated LoRA for each task individually. Additionally, conducting an ablation study on the impact of top-k would be informative.

**Questions:**

How can you attain top-2 sparsity gates while ensuring compatibility with gradient flow?

---

> ### Author Response · Authors · 2023-11-16
>
> > ***1. Given our awareness of each example's task, an important baseline involves employing a dedicated LoRA for each task individually. Additionally, conducting an ablation study on the impact of top-k would be informative.***
>
> **Table A**
> |method|VOC Recall@ 0.5|ScienceQA Acc@1|Flickr30k CIDEr|CIFAR-10 Acc@1|FacialAttr Smile Acc@1|
> |:----:|:----:|:----:|:----:|:----:|:----:|
> |LAMM (LoRA)|7.61|40.31|0.21|73.50|50.15|
> |LAMM (LoRA-MoE)|39.04|46.95|5.66|65.40|59.82|
> |LAMM (LoRA-individual)|28.38|48.54|5.83|82.11|*28.76|
>
> * We provide results in **Table A**. Specifically, we classify the 2D visual tasks as 6 types, i.e., QA, description, classification, detection, n-round conversations, and others, in instruction datasets, and employ 6 LoRAs for each task individually for joint training. During evaluation, we manually assign the expert corresponding to the most similar tasks in training to the downstream tasks.
>
> * The problem with employing separate LoRA for each task is that **it is difficult to assign a suitable expert for downstream tasks that are not present in instruction datasets**, e.g., Facial Attribute Recognition. In this task, we attempted to assign "others" or "description" experts to it, but both results were not good (see * in the table, we selected the "others" expert because it is relatively better). Moreover, **manually inputting the task type or the selected expert into the model during inference is not as efficient as the automatic gating based on questions in LoRA-MoE**.
>
> * Besides, **if we have no idea what downstream tasks will be performed, it is challenging to design a LoRA for each training task due to potential generalization issues**.
>
> > ***2. Additionally, conducting an ablation study on the impact of top-k would be informative.***
>
> **Table B**
> |Gate|#Experts|VOC Recall@ 0.5|VOC Prec@ 0.5|ScienceQA Acc@1|
> |:----:|:----:|:----:|:----:|:----:|
> |-|1|7.61|5.95|40.31|
> |top-1|4|22.42|21.23|36.88|
> |top-2|4|39.04|35.21|46.95|
> |top-3|4|38.57|36.02|43.89|
>
> * We provide results of adopting different top-k gates in **Table B**. We will add this table in the final version.
>
> * From the table, we can find:
>   * Top-1 gate has a poor performance due to its less flexibility in choice (4 combinations) compared with top-2/top-3 gate.
>   * While in the top-3 gate scenario, we find that the third choice is relatively small in most tasks (e.g., in classification tasks, the routing weight is [0.1, 0, 0.4, 0.5]), therefore it is very similar to the top-2 gate, and achieves similar results as the top-2 gate.
>
> > ***3. How can you attain top-2 sparsity gates while ensuring compatibility with gradient flow?***
>
> * We implement the top-2 gate based on an open-source implementation of [1] from [repo1](https://github.com/lucidrains/mixture-of-experts/tree/master) and [repo2](https://github.com/davidmrau/mixture-of-experts). We can give you a simple PyTorch-like pseudo code for top-2 gate implementation.
>
>   ```python
>   class Top2Gate(nn.Module):
>       def __init__(self, num_experts, ...):
>           self.num_experts = num_experts
>           self.w_gating = nn.Parameter(torch.randn(input_dim, num_experts))
>           ...
>
>       @staticmethod
>       def top1(tensor):
>           return tensor.topk(k=1, dim=-1)
>
>       def forward(self, x):  # x: bs, n_token, dim
>           gate = torch.einsum('bnd,de->bne', x, self.w_gating)  # gate: bs, n_token, n_experts
>           gate = gate.mean(dim=-1).softmax(dim=-1)  # reduce token, gate: bs, n_experts
>
>           gate_1, index_1 = self.top1(gate)
>           mask_1 = F.one_hot(index_1, self.num_experts)
>
>           gate_without_top1 = gate * (1. - mask_1)
>           gate_2, index_2 = self.top1(gate_without_top1)
>
>           norm = gate_1 + gate_2 + eps
>           gate_1 /= norm
>           gate_2 /= norm
>
>           top2_gate = torch.zeros(x.shape[0], self.num_experts)  # top2_gate: bs, n_experts
>           top2_gate.scatter_(1, index_1.view(-1, 1), gate_1.view(-1, 1))
>           top2_gate.scatter_(1, index_2.view(-1, 1), gate_2.view(-1, 1))
>           return top2_gate
>   ```
>
> * Other implementation details:
>
>   * Given the input question embedding and N LoRA experts, we can obtain N-dim routing weights using the top-2 gate, where two experts with the highest scores are retained and the others are set to zero. Next, we can calculate the weighted sum of N LoRA experts. The non-activated experts are multiplied by a coefficient of 0 in routing weights.
>
>   * This approach has almost no significant impact on the computation cost when N=4/6 in our setting during fine-tuning. During inference, non-activated experts can be ignored, thus improving efficiency. Furthermore, if the tasks and questions are specified, LoRA-MoE can merge parameter weights with LLM like vanilla LoRA to reduce extra inference costs.
>
> [1] Outrageously Large Neural Networks: The Sparsely-Gated Mixture-of-Experts Layer. 2017, Shazeer et al.

---

> > ### Author Response · Authors · 2023-11-22
> >
> > We hope this message finds you well. We have noted the deadline for open discussion of ICLR 2024 is approaching, yet we have not yet received any feedback from you. In light of this, we sincerely wish to know if we can receive any updated comments regarding to our submission 1570, titled "Octavius: Mitigating Task Interference in MLLMs via MoE". We are very pleased to hear from you on the reviewer’s comments.

---

### Meta-Review · Area_Chair_m1Ch · 2023-11-29

**Metareview:**

The paper proposes leveraging LoRA to build mixture of experts models at finetuning stage, to mitigate tug-of-war issues (finetuning foundation models on mutliple tasks / multiple modalities yields suboptimal performance compared to single task/modality used in finetuning) during finetuning. A sparse gate routing scheme is used to learn LoRA experts separately and dynamically to mitigate knowledge interference across tasks/modalities. Empirical results on 2D and 3D downstream tasks show a consistent performance gain is achieved with proposed approach compared to baselines such as like LAMM and LLaVA-LoRA.

Strength:
 - The work is simple and seems generalizable to various foundation models, tasks, modalities.
 - The work focuses on a popular areas (finetuning foundation models) and yields non-trivial performance gain, which make the work of reasonable interests to ICLR community.
 - It's a reasonable idea to apply MoE and LoRA to distribute the different types of knowledge and improve multi-task learning, which makes the (generalizability of the) result more convincing.
 - Reasonable design of experiments and ablation study supports authors' claim and effectiveness of proposed approach.

Weakness:
 - Some concerns about scalability of the approach. This paper only focuses on supervised finetuning and using few foundation models as baseline, which is understandable given limited page length and authors' bandwidth. However, it's natural for the community to wonder what if the approach is applied to pre-training, to a lot more experts/tasks/modalities, or to a much bigger / more performant foundation models. Would the gain diminishing because more generic knowledge is learned in the foundation models? The community would definitely love to see these questions answered, perhaps over multiple research studies, to determine the effectiveness of proposed method (or MoE in general).

**Justification For Why Not Higher Score:**

as mentioned in the weakness, the experiment results are not strong enough to show undoubted effectiveness of proposed approach. thus I wouldn't suggest to put the work in oral or spotlight.

**Justification For Why Not Lower Score:**

the work is novel and working on areas (foundation model finetuning) of general interests in ICLR community. thus i would recoomend to accept this work.

---

### Decision · Program_Chairs · 2024-01-16

Accept (poster)